# Predicting non-linear dynamics by stable local learning in a recurrent spiking neural network

**Aditya Gilra[1,2]\*, Wulfram Gerstner[1,2]**

[1]Brain-Mind Institute, School of Life Sciences, École Polytechnique Fédérale de Lausanne, Lausanne, Switzerland; [2]School of Computer and Communication Sciences, École Polytechnique Fédérale de Lausanne, Lausanne, Switzerland

**Abstract** The brain needs to predict how the body reacts to motor commands, but how a network of spiking neurons can learn non-linear body dynamics using local, online and stable learning rules is unclear. Here, we present a supervised learning scheme for the feedforward and recurrent connections in a network of heterogeneous spiking neurons. The error in the output is fed back through fixed random connections with a negative gain, causing the network to follow the desired dynamics. The rule for Feedback-based Online Local Learning Of Weights (FOLLOW) is local in the sense that weight changes depend on the presynaptic activity and the error signal projected onto the postsynaptic neuron. We provide examples of learning linear, non-linear and chaotic dynamics, as well as the dynamics of a two-link arm. Under reasonable approximations, we show, using the Lyapunov method, that FOLLOW learning is uniformly stable, with the error going to zero asymptotically.

DOI: https://doi.org/10.7554/eLife.28295.001

## Introduction

Over the course of life, we learn many motor tasks such as holding a pen, chopping vegetables, riding a bike or playing tennis. To control and plan such movements, the brain must implicitly or explicitly learn forward models (*Conant and Ross Ashby, 1970*) that predict how our body responds to neural activity in brain areas known to be involved in motor control (*Figure 1A*). More precisely, the brain must acquire a representation of the dynamical system formed by our muscles, our body, and the outside world in a format that can be used to plan movements and initiate corrective actions if the desired motor output is not achieved (*Pouget and Snyder, 2000*; *Wolpert and Ghahramani, 2000*; *Lalazar and Vaadia, 2008*). Visual and/or proprioceptive feedback from spontaneous movements during pre-natal (*Khazipov et al., 2004*) and post-natal development (*Petersson et al., 2003*) or from voluntary movements during adulthood (*Wong et al., 2012*; *Hilber and Caston, 2001*) are important to learn how the body moves in response to neural motor commands (*Lalazar and Vaadia, 2008*; *Wong et al., 2012*; *Sarlegna and Sainburg, 2009*; *Dadarlat et al., 2015*), and how the world reacts to these movements (*Davidson and Wolpert, 2005*; *Zago et al., 2005*, *2009*; *Friston, 2008*). We wondered whether a non-linear dynamical system, such as a forward predictive model of a simplified arm, can be learned and represented in a heterogeneous network of spiking neurons by adjusting the weights of recurrent connections.

Supervised learning of recurrent weights to predict or generate non-linear dynamics, given command input, is known to be difficult in networks of rate units, and even more so in networks of spiking neurons (*Abbott et al., 2016*). Ideally, in order to be biologically plausible, a learning rule must be *online* that is constantly incorporating new data, as opposed to batch learning where weights are adjusted only after many examples have been seen; and *local* that is the quantities that modify the

**\*For correspondence:**
aditya.gilra@epfl.ch

**Competing interests:** The authors declare that no competing interests exist.

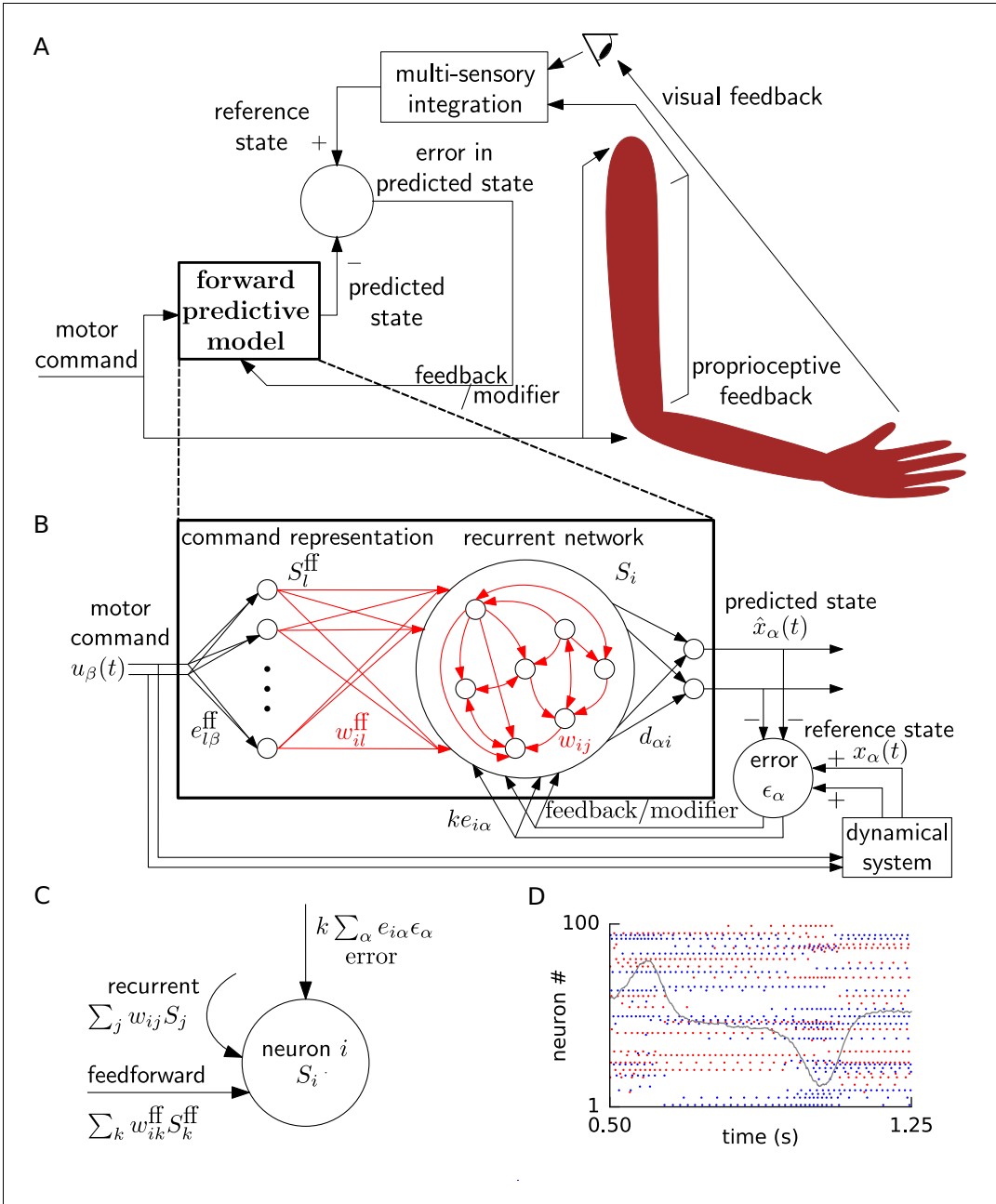

**Figure 1.** Schematic for learning a forward model. (**A**) During learning, random motor commands (motor babbling) cause movements of the arm, and are also sent to the forward predictive model, which must learn to predict the joint angles and velocities (state variables) of the arm. The deviation of the predicted state from the reference state, obtained by visual and proprioceptive feedback, is used to learn the forward predictive model with architecture shown in B. (**B**) Motor command $\vec{u}$ is projected onto neurons with random weights $e_{l\beta}^{\text{ff}}$. The spike trains of these command representation neurons $S_l^{\text{ff}}$ are sent via plastic feedforward weights $w_{il}^{\text{ff}}$ into the neurons of the recurrent network having plastic weights $w_{ij}$ (plastic weights in red). Readout weights $d_{\alpha i}$ decode the filtered spiking activity of the recurrent network as the predicted state $\hat{x}_\alpha(t)$. The deviation of the predicted state from the reference state of the reference dynamical system in response to the motor command, is fed back into the recurrent network with error encoding weights $ke_{i\alpha}$. (**C**) A cartoon depiction of feedforward, recurrent and error currents entering a neuron $i$ in the recurrent network. (**D**) Spike trains of a few randomly selected neurons of the recurrent network from the non-linear oscillator example are plotted (alternate red and blue colours are for guidance of eye only). A component $\hat{x}_2$ of the network output during a period of the oscillator is overlaid on the spike trains to indicate their relation to the output.

*Figure 1 continued on next page*

*Figure 1 continued*

DOI: https://doi.org/10.7554/eLife.28295.002

The following figure supplement is available for figure 1:

**Figure supplement 1.** Gain functions of heterogeneous neurons.

DOI: https://doi.org/10.7554/eLife.28295.003

weight of a synapse must be available locally at the synapse as opposed to backpropagation through time (BPTT) (*Rumelhart et al., 1986*) or real-time recurrent learning (RTRL) (*Williams and Zipser, 1989*) which are non-local in time or in space, respectively (*Pearlmutter, 1995*; *Jaeger, 2005*). Even though Long-Short-Term-Memory (LSTM) units (*Hochreiter and Schmidhuber, 1997*) avoid the vanishing gradient problem (*Bengio et al., 1994*; *Hochreiter et al., 2001*) in recurrent networks, the corresponding learning rules are difficult to interpret biologically.

Our approach toward learning of recurrent spiking networks is situated at the crossroads of reservoir computing (*Jaeger, 2001*; *Maass et al., 2002*; *Legenstein et al., 2003*; *Maass and Markram, 2004*; *Jaeger and Haas, 2004*; *Joshi and Maass, 2005*; *Legenstein and Maass, 2007*), FORCE learning (*Sussillo and Abbott, 2009*, *2012*; *DePasquale et al., 2016*; *Thalmeier et al., 2016*; *Nicola and Clopath, 2016*), function and dynamics approximation (*Funahashi, 1989*; *Hornik et al., 1989*; *Girosi and Poggio, 1990*; *Sanner and Slotine, 1992*; *Funahashi and Nakamura, 1993*; *Pouget and Sejnowski, 1997*; *Chow and Xiao-Dong Li, 2000*; *Seung et al., 2000*; *Eliasmith and Anderson, 2004*; *Eliasmith, 2005*) and adaptive control theory (*Morse, 1980*; *Narendra et al., 1980*; *Slotine and Coetsee, 1986*; *Weiping Li et al., 1987*; *Narendra and Annaswamy, 1989*; *Sastry and Bodson, 1989*; *Ioannou and Sun, 2012*). In contrast to the original reservoir scheme (*Jaeger, 2001*; *Maass et al., 2002*) where learning was restricted to the readout connections, we focus on a learning rule for the recurrent connections. Whereas neural network implementations of control theory (*Sanner and Slotine, 1992*; *DeWolf et al., 2016*) modified adaptive feedback weights without a synaptically local interpretation, we modify the recurrent weights in a synaptically local manner. Compared to FORCE learning where recurrent synaptic weights have to change rapidly during the initial phase of learning (*Sussillo and Abbott, 2009*, *2012*), we aim for a learning rule that works in the biologically more plausible setting of slow synaptic changes. While previous work has shown that linear dynamical systems can be represented and learned with local online rules in recurrent spiking networks (*MacNeil and Eliasmith, 2011*; *Bourdoukan and Denève, 2015*), for non-linear dynamical systems the recurrent weights in spiking networks have typically been computed offline (*Eliasmith, 2005*).

Here, we propose a scheme for how a recurrently connected network of heterogeneous deterministic spiking neurons may learn to mimic a low-dimensional non-linear dynamical system, with a local and online learning rule. The proposed learning rule is supervised, and requires access to the error in observable outputs. The output errors are fed back with random, but fixed feedback weights. Given a set of fixed error-feedback weights, the learning rule is synaptically local and combines presynaptic activity with the local postsynaptic error variable.

## Results

A forward predictive model (*Figure 1A*) takes, at each time step, a motor command $\vec{u}(t)$ as input and predicts the next observable state $\vec{\hat{x}}(t + \Delta t)$ of the system. In the numerical implementation, we consider $\Delta t = 1\text{ms}$, but for the sake of notational simplicity we drop the $\Delta t$ in the following. The predicted system state $\vec{\hat{x}}$ (e.g., the vector of joint angles and velocities of the arm) is assumed to be low-dimensional with dimensionality $N_d$ (4-dimensional for a two-link arm). The motor command $\vec{u}(t)$ is used to generate target movements such as 'lift your arm to a location', with a dimensionality $N_c$ of the command typically smaller than the dimensionality $N_d$ of the system state.

The actual state of the reference system (e.g., actual joint angles and velocities of the arm) is described by a non-linear dynamical system, which receives the control input $\vec{u}(t) \in \mathbb{R}^{N_c}$ and evolves according to a set of coupled differential equations

$$\frac{dx_\alpha(t)}{dt} = h_\alpha(\vec{x}(t), \vec{u}(t)), \tag{1}$$

where $\vec{x}$ with components $x_\alpha$ (where $\alpha = 1, \ldots, N_d$) is the vector of observable state variables, and $\vec{h}$ is a vector whose components are arbitrary non-linear functions $h_\alpha$. For example, the observable system state $\vec{x}(t)$ could be the joint angles and velocities of the arm deduced from visual and proprioceptive input (*Figure 1A*). We show that, with training, the forward predictive model learns to make the error

$$\epsilon_\alpha \equiv x_\alpha(t) - \hat{x}_\alpha(t) \tag{2}$$

between the actual state $\vec{x}(t)$ and the predicted state $\hat{\vec{x}}(t)$ negligible.

## Network architecture for learning the forward predictive model

In our neural network model (*Figure 1B*), the motor command $\vec{u}(t)$ drives the spiking activity of a command representation layer of 3000 to 5000 leaky integrate-and-fire neurons via connections with fixed random weights. These neurons project, via plastic feedforward connections, to a recurrent network of also 3000 to 5000 integrate-and-fire neurons. We assume that the predicted state $\hat{\vec{x}}$ is linearly decoded from the activity of the recurrent network. Denoting the spike train of neuron $i$ by $S_i(t)$, the component $\alpha$ of the predicted system state is

$$\hat{x}_\alpha(t) = \sum_i d_{\alpha i} \int_{-\infty}^t S_i(s) \kappa(t-s) ds \equiv \sum_i d_{\alpha i} (S_i * \kappa)(t), \tag{3}$$

where $d_{\alpha i}$ are the readout weights. The integral represents a convolution with a low-pass filter

$$\kappa(t) \equiv \exp(-t/\tau_s)/\tau_s, \tag{4}$$

with a time constant $\tau_s = 20$ ms, and is denoted by $(S_i * \kappa)(t)$.

The current into a neuron with index $l$ ($l = 1, \ldots, N$), in the command representation layer comprising $N$ neurons, is

$$J_l^{\mathrm{ff}} = \sum_\alpha e_{l\alpha}^{\mathrm{ff}} u_\alpha + b_l^{\mathrm{ff}}, \tag{5}$$

where $e_{l\alpha}^{\mathrm{ff}}$ are fixed random weights, while $b_l^{\mathrm{ff}}$ is a neuron-specific constant for bias (see Methods) (*Eliasmith and Anderson, 2004*). We use Greek letters for the indices of low-dimensional variables (such as command) and Latin letters for neuronal indices, with summations going over the full range of the indices. The number of neurons $N$ in the command representation layer is much larger than the dimensionality of the input, that is $N \gg N_c$.

The input current to a neuron with index $i$ ($i = 1, \ldots, N$) in the recurrent network is

$$J_i = \sum_l w_{il}^{\mathrm{ff}} (S_l^{\mathrm{ff}} * \kappa)(t) + \sum_j w_{ij} (S_j * \kappa)(t) + \sum_\alpha k e_{i\alpha} (\epsilon_\alpha * \kappa)(t) + b_i, \tag{6}$$

where $w_{il}^{\mathrm{ff}}$ and $w_{ij}$ are the feedforward and recurrent weights, respectively, which are both subject to our synaptic learning rule, whereas $k e_{i\alpha}$ are fixed error feedback weights (see below). The spike trains travelling along the feedforward path $S_l^{\mathrm{ff}}$ and those within the recurrent network $S_j$ are both low-pass filtered (convolution denoted by $*$) at the synapses with the exponential filter $\kappa$ defined above. The constant parameter $b_i$ is a neuron specific bias (see Methods). The constant $k>0$ is the gain for feeding back the output error. The number of neurons $N$ in the recurrent network is much larger than the dimensionality $N_d$ of the represented variable $\hat{x}$, that is $N \gg N_d$.

For all numerical simulations, we used deterministic leaky integrate and fire (LIF) neurons. The voltage $V_l$ of each LIF neuron indexed by $l$, was a low-pass filter of its driving current $J_l$:

$$\tau_m \frac{dV_l}{dt} = -V_l + J_l, \tag{7}$$

with a membrane time constant, of $\tau_m = 20$ ms. The neuron fired when the voltage $V_l$ crossed a threshold $\theta = 1$ from below, after which the voltage was reset to zero for a refractory period $\tau_r$ of 2 ms. If the voltage went below zero, it was clipped to zero. Mathematically, the spike trains $S_l^{\mathrm{ff}}(t)$ in

the command representation layer and $S_l(t)$ in the recurrent network, are a sequence of events, modelled as a sum of Dirac delta-functions.

Biases and input weights of the spiking neurons vary between one neuron and the next, both in the command representation layer and the recurrent network, yielding different frequency versus input curves for different neurons (*Figure 1—figure supplement 1*). Since arbitrary low-dimensional functions can be approximated by linear decoding from a basis of non-linear functions (*Funahashi, 1989*; *Girosi and Poggio, 1990*; *Hornik et al., 1989*), such as neuronal tuning curves (*Sanner and Slotine, 1992*; *Seung et al., 2000*; *Eliasmith and Anderson, 2004*), we may expect that suitable feedforward weights onto, and lateral weights within, the recurrent network can be found that approximate the role of the function $\vec{h}$ in *Equation (1)*. In the next subsection, we propose an error feedback architecture along with a local and online synaptic plasticity rule that can train these feedforward and recurrent weights to approximate this role, while the readout weights are kept fixed, so that the network output mimics the dynamics in *Equation (1)*.

## Negative error feedback via auto-encoder enables local learning

To enable weight tuning, we make four assumptions regarding the network architecture. The initial two assumptions are related to input and output. First, we assume that, during the learning phase, a random time-dependent motor command input $\vec{u}(t)$ is given to both the muscle-body reference system described by *Equation (1)* and to the spiking network. The random input generates irregular trajectories in the observable state variables, mimicking motor babbling (*Meltzoff and Moore, 1997*; *Petersson et al., 2003*). Second, we assume that each component $\hat{x}_\alpha$ of the output predicted by the spiking network is compared to the actual observable output $x_\alpha$ produced by the reference system of *Equation (1)* and their difference (the output error $\epsilon_\alpha$; *Equation (2)*) is calculated, similar to supervised learning schemes such as perceptron learning (*Rosenblatt, 1961*).

The final two assumptions are related to the error feedback. Our third assumption is that the readout weights $d_{\alpha i}$ have been pre-learned, possibly earlier in development, in the absence of feedforward and recurrent connections, so as to form an auto-encoder of gain $k$ with the fixed random feedback weights $ke_{i\alpha}$. Specifically, an arbitrary value $\epsilon_\alpha$ sent via the error feedback weights to the recurrent network and read out, from its $N$ neurons, via the decoding weights gives back (approximately) $k\epsilon_\alpha$. Thus, we set the decoding weights so as to minimize the squared error between the decoded output and required output $k\vec{\epsilon}$ for a set of randomly chosen vectors $\vec{\epsilon}$ while setting feedforward and recurrent weights to zero (see Methods). We used an algorithmic learning scheme here, but we expect that these decoding weights can also be pre-learned by biologically plausible learning schemes (*D'Souza et al., 2010*; *Urbanczik and Senn, 2014*; *Burbank, 2015*).

Fourth, we assumed that the error $\epsilon_\alpha = x_\alpha - \hat{x}_\alpha$ is projected back to neurons in the recurrent network through the above-mentioned fixed random feedback weights. From the third term in *Equation (6)* and *Figure 1B–C*, we define a total error input that neuron $i$ receives:

$$I_i^\epsilon \equiv k \sum_\alpha e_{i\alpha} \epsilon_\alpha, \tag{8}$$

with feedback weights $ke_{i\alpha}$, where $k$ is fixed at a large constant positive value.

The combination of the auto-encoder and the error feedback implies that the output stays close to the reference, as explained now. In open loop that is without connecting the output $\vec{\hat{x}}$ and the reference $\vec{x}$ to the error node, an input $\vec{\epsilon}$ to the network generates an output $\vec{\hat{x}} = k\vec{\epsilon}$ due to the auto-encoder of gain $k$. In closed loop, that is with the output and reference connected to the error node (*Figure 1B*), the error input is $\vec{\epsilon} = \vec{x} - \vec{\hat{x}}$, and the network output $\vec{\hat{x}}$ settles to:

$$\vec{\hat{x}} = k\vec{\epsilon} = k\left(\vec{x} - \vec{\hat{x}}\right)$$
$$\implies \vec{\hat{x}} = \frac{k}{k+1}\vec{x} \approx \vec{x}, \tag{9}$$

that is approximately the reference $\vec{x}$ for large positive $k$. The fed-back residual error $\vec{\epsilon} = \vec{x}/(k+1)$ drives the neural activities and thence the network output. Thus, feedback of the error causes the output $\hat{x}_\alpha$ to approximately follow $x_\alpha$, for each component $\alpha$, as long as the error feedback time scale is fast compared to the reference dynamical system time scale, analogous to negative error feedback in adaptive control (*Narendra and Annaswamy, 1989*; *Ioannou and Sun, 2012*).

While error feedback is on, the synaptic weights $w_{il}^{\mathrm{ff}}$ and $w_{ij}$ on the feedforward and recurrent connections, respectively, are updated as:

$$\begin{aligned}
\dot{w}_{il}^{\mathrm{ff}} &= \eta\,(I_i^\epsilon * \kappa^\epsilon)(S_l^{\mathrm{ff}} * \kappa)(t),\\
\dot{w}_{ij} &= \eta\,(I_i^\epsilon * \kappa^\epsilon)(S_j * \kappa)(t),
\end{aligned} \tag{10}$$

where $\eta$ is the learning rate (which is either fixed or changes on the slow time scale of minutes), and $\kappa^\epsilon$ is an exponentially decaying filter kernel with a time constant of 80 or 200 ms. For a postsynaptic neuron $i$, the error term $I_i^\epsilon * \kappa^\epsilon$ is the same for all its synapses, while the presynaptic contribution is synapse-specific.

We call the learning scheme 'Feedback-based Online Local Learning Of Weights' (FOLLOW), since the predicted state $\vec{\hat{x}}$ *follow*s the true state $\vec{x}$ from the start of learning. Under precise mathematical conditions, we show in the Methods that the FOLLOW scheme converges to a stable solution, while simultaneously deriving the learning rule.

Because of the error feedback, with constant $k \gg 1$, the output is close to the reference from the start of learning. However, initially the error is not exactly zero, and this non-zero error drives the weight updates via *Equation (10)*. After a sufficiently long learning time, a vanishing error ($\epsilon_\alpha = 0$ for all components) indicates that the neuronal network now autonomously generates the desired output, so that feedback is no longer required. In the Methods section, we show that not just the low-dimensional output $\vec{\hat{x}}$, but also the spike trains $S_i(t)$, for $i = 1, \ldots, N$, are entrained by the error feedback to be close to the ideal ones required to generate $\vec{x}$.

During learning, the error feedback via the auto-encoder in a loop serves two roles: (i) to make the error current available in each neuron, projected correctly, for a local synaptic plasticity rule, and (ii) to drive the spike trains to the target ones for producing the reference output. In other learning schemes for recurrent neural networks, where neural activities are not constrained by error feedback, it is difficult to assign credit or blame for the momentarily observed error, because neural activities from the past affect the present output in a recurrent network. In the FOLLOW scheme, the spike trains are constrained to closely follow the ideal time course throughout learning, so that the present error can be attributed directly to the weights, enabling us to change the weights with a simple perceptron-like learning rule (*Rosenblatt, 1961*) as in *Equation (10)*, bypassing the credit assignment problem. In the perceptron rule, the weight change $\Delta w \sim (\mathrm{pre}) \cdot \delta$ is proportional to the presynaptic input $(\mathrm{pre})$ and the error $\delta$. In the FOLLOW learning rule of *Equation (10)*, we can identify $(S_i * \kappa)$ with $(\mathrm{pre})$ and $(I_i^\epsilon * \kappa^\epsilon)$ with $\delta$. In Methods, we derive the learning rule of *Equation (10)* in a principled way from a stability criterion.

FORCE learning (*Sussillo and Abbott, 2009*, *2012*; *DePasquale et al., 2016*; *Thalmeier et al., 2016*; *Nicola and Clopath, 2016*) also clamps the output and neural activities to be close to ideal during learning, by using weight changes that are faster than the time scale of the dynamics. In our FOLLOW scheme, clamping is achieved via negative error feedback using the auto-encoder, which allows weight changes to be slow and makes the error current available locally in the post-synaptic neuron. Other methods used feedback based on adaptive control for learning in recurrent networks of spiking neurons, but were limited to linear systems (*MacNeil and Eliasmith, 2011*; *Bourdoukan and Denève, 2015*), whereas the FOLLOW scheme was derived for non-linear systems (see Methods). Our learning rule of *Equation (10)* uses an error $\epsilon_\alpha \equiv x_\alpha - \hat{x}_\alpha$ in the observable state, rather than an error involving the derivative $dx_\alpha/dt$ in *Equation (1)*, as in other schemes (see Appendix 1) (*Eliasmith, 2005*; *MacNeil and Eliasmith, 2011*). The reader is referred to Discussion for detailed further comparisons. The FOLLOW learning rule is local since all quantities needed on the right-hand-side of *Equation (10)* could be available at the location of the synapse in the postsynaptic neuron. For a potential implementation and prediction for error-based synaptic plasticity, and for a critical evaluation of the notion of 'local rule', we refer to the Discussion.

## Spiking networks learn target dynamics via FOLLOW learning

In order to check whether the FOLLOW scheme would enable the network to learn various dynamical systems, we studied three systems describing a non-linear oscillator (*Figure 2*), low-dimensional chaos (*Figure 3*) and simulated arm movements (*Figure 4*) (additional examples in *Figure 2—figure supplement 2*, *Figure 2—figure supplement 4* and Methods). In all simulations, we started with vanishingly small feedforward and recurrent weights (tabula rasa), but assumed pre-learned readout

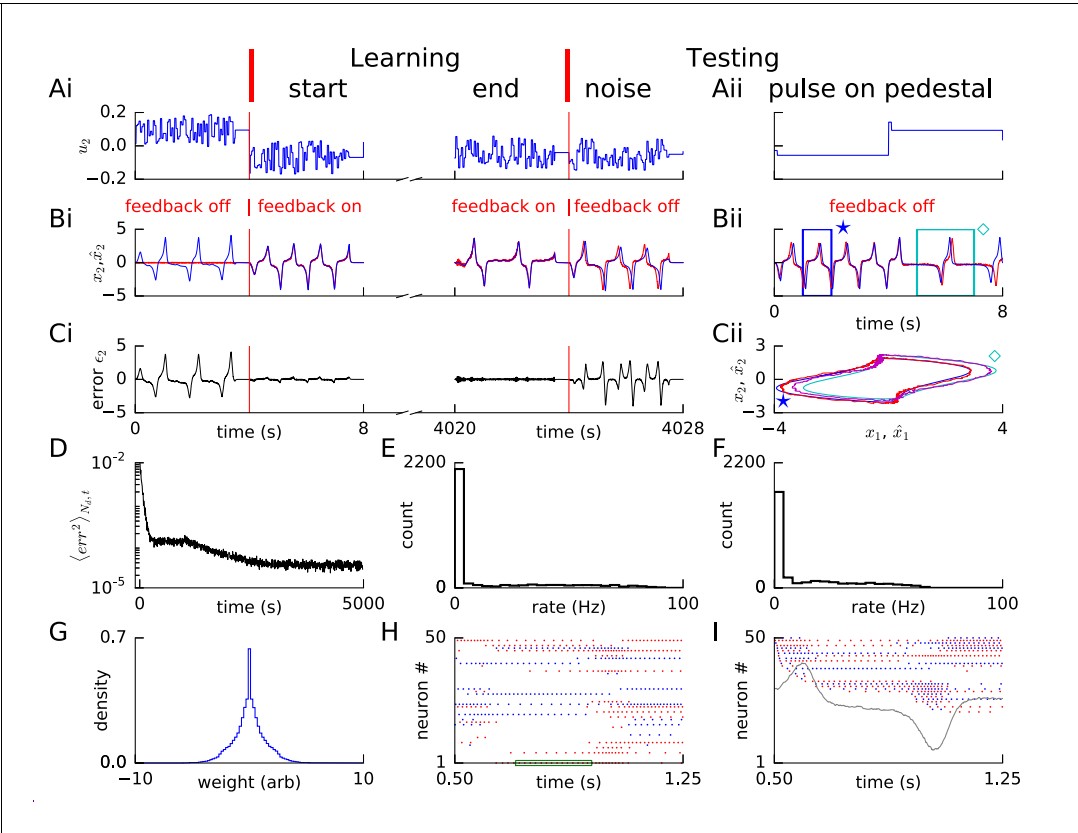

**Figure 2.** Learning non-linear dynamics via FOLLOW: the van der Pol oscillator. (A-C) Control input, output, and error are plotted versus time: before the start of learning; in the first 4 s and last 4 s of learning; and during testing without error feedback (demarcated by the vertical red lines). Weight updating and error current feedback were both turned on after the vertical red line on the left at the start of learning, and turned off after the vertical red line in the middle at the end of learning. (A) Second component of the input $u_2$. (B) Second component of the learned dynamical variable $\hat{x}_2$ (red) decoded from the network, and the reference $x_2$ (blue). After the feedback was turned on, the output tracked the reference. The output continued to track the reference approximately, even after the end of the learning phase, when feedback and learning were turned off. The output tracked the reference approximately, even with a very different input (Bii). With higher firing rates, the tracking without feedback improved (*Figure 2—figure supplement 1*). (C) Second component of the error $\epsilon_2 = x_2 - \hat{x}_2$ between the reference and the output. (Cii) Trajectory $(x_1(t), x_2(t))$ in the phase plane for reference (red,magenta) and prediction (blue,cyan) during two different intervals as indicated by ★ and ◇ in Bii. (D) Mean squared error per dimension averaged over 4 s blocks, on a log scale, during learning with feedback on. Learning rate was increased by a factor of 20 after 1,000 s to speed up learning (as seen by the sharp drop in error at 1000 s). (E) Histogram of firing rates of neurons in the recurrent network averaged over 0.25 s (interval marked in green in H) when output was fairly constant (mean across neurons was 12.4 Hz). (F) As in E, but averaged over 16 s (mean across neurons was 12.9 Hz). (G) Histogram of weights after learning. A few strong weights $|w_{ij}|>10$ are out of bounds and not shown here. (H) Spike trains of 50 randomly-chosen neurons in the recurrent network (alternating colors for guidance of eye only). (I) Spike trains of H, reverse-sorted by first spike time after 0.5 s, with output component $\hat{x}_2$ overlaid for timing comparison.

DOI: https://doi.org/10.7554/eLife.28295.004

The following figure supplements are available for figure 2:

**Figure supplement 1.** Learning van der Pol oscillator dynamics via FOLLOW with higher firing rates.

DOI: https://doi.org/10.7554/eLife.28295.005

**Figure supplement 2.** Learning linear dynamics via FOLLOW: 2D decaying oscillator.

DOI: https://doi.org/10.7554/eLife.28295.006

**Figure supplement 3.** Readout weights learn if recurrent weights are as is, but not if shuffled.

DOI: https://doi.org/10.7554/eLife.28295.007

**Figure supplement 4.** Learning non-linear feedforward transformation with linear recurrent dynamics via FOLLOW.

DOI: https://doi.org/10.7554/eLife.28295.008

**Figure supplement 5.** Feedforward weights are uncorrelated, while recurrent ones are correlated, when learning same recurrent dynamics but with different feedforward transforms.

DOI: https://doi.org/10.7554/eLife.28295.009

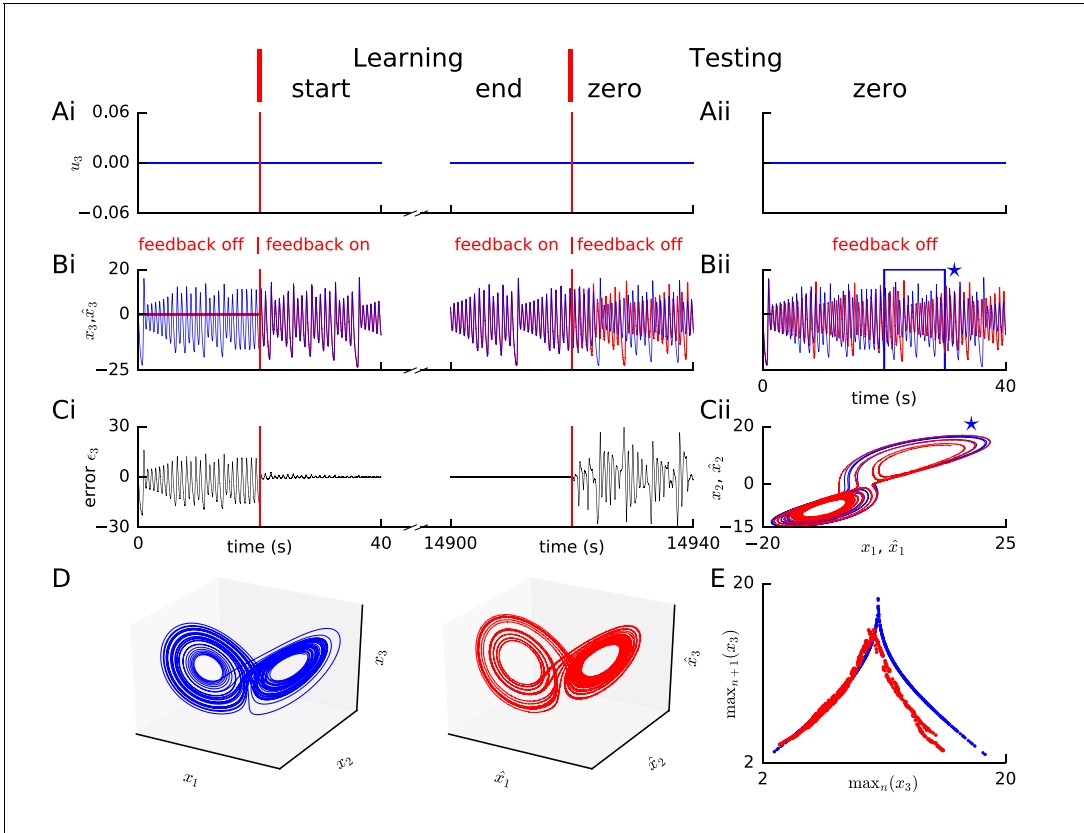

**Figure 3.** Learning chaotic dynamics via FOLLOW: the Lorenz system. Layout and legend of panels (A-C) are analogous to *Figure 2A–C*. (D) The trajectories of the reference (left panel) and the learned network (right panel) are shown in state space for 40 s with zero input during the testing phase, forming the well-known Lorenz attractor. (E) Tent map, that is local maximum of the third component of the reference signal (blue)/network output (red) is plotted versus the previous local maximum, for 800 s of testing with zero input. The reference is plotted with filtering in panels (A-C), but unfiltered for the strange attractor (panel D left) and the tent map (panel E blue).

DOI: https://doi.org/10.7554/eLife.28295.010

The following figure supplement is available for figure 3:

**Figure supplement 1.** Learning the Lorenz system without filtering the reference variables.

DOI: https://doi.org/10.7554/eLife.28295.011

weights matched to the error feedback weights. For each of the three dynamical systems, we had a learning phase and a testing phase. During each phase, we provided time-varying input to both the network (*Figure 1B*) and the reference system. During the learning phase, rapidly changing control signals mimicked spontaneous movements (motor babbling) while synaptic weights were updated according to the FOLLOW learning rule *Equation (10)*.

During learning, the mean squared error, where the mean was taken over the number of dynamical dimensions $N_d$ and over a duration of a few seconds, decreased (*Figure 2D*). We stopped the learning phase that is weight updating, when the mean squared error approximately plateaued as a function of learning time (*Figure 2D*). At the end of the learning phase, we switched the error feedback off ('open loop') and provided different test inputs to both the reference system and the recurrent spiking network. A successful forward predictive model should be able to predict the state variables in the open-loop model over a finite time horizon (corresponding to the planning horizon of a short action sequence) and in the closed-loop mode (with error feedback) without time limit.

## Non-linear oscillator

Our FOLLOW learning scheme enabled a network with 3000 neurons in the recurrent network and 3000 neurons in the motor command representation layer to approximate the non-linear 2-dimensional van der Pol oscillator (*Figure 2*). We used a superposition of random steps as input, with

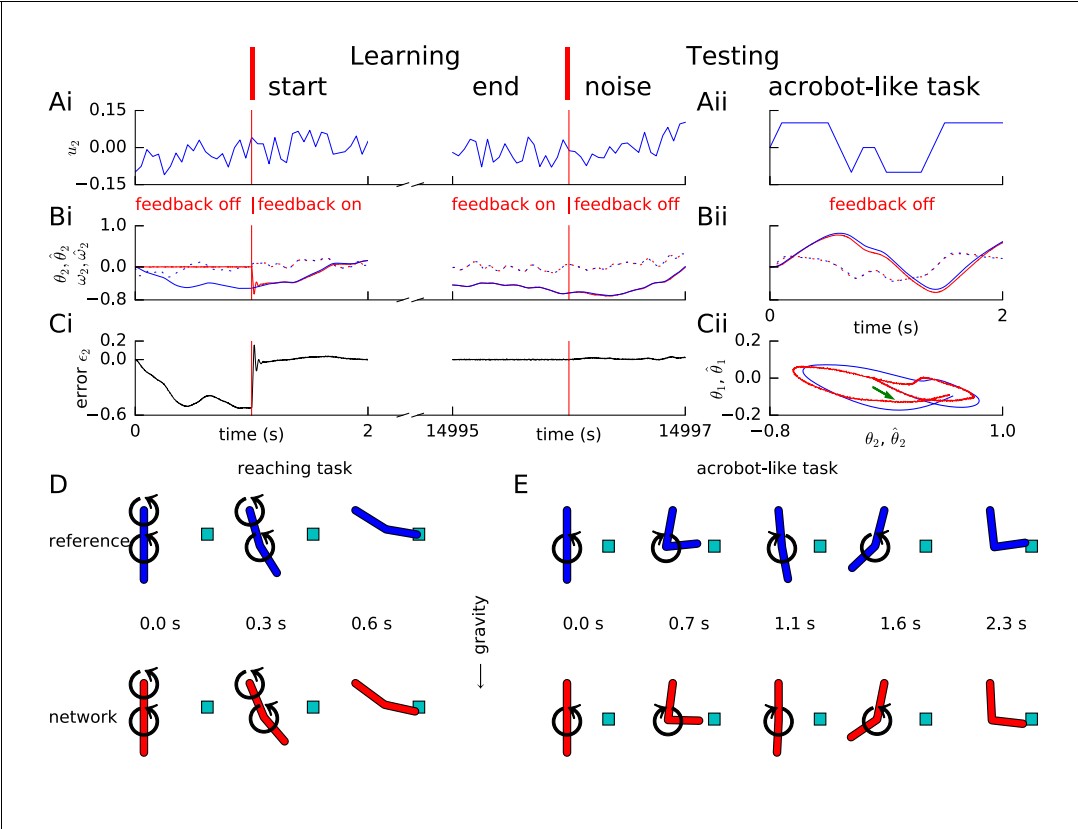

**Figure 4.** Learning arm dynamics via FOLLOW. Layout and legend of panels A-C are analogous to *Figure 2A–C* except that: in panel (**A**), the control input (torque) on the elbow joint is plotted; in panel (**B**), reference and decoded angle $\theta_2, \hat{\theta}_2$ (solid) and angular velocity $\omega_2, \hat{\omega}_2$ (dotted) are plotted, for the elbow joint; in panel (**C**), the error $\theta_2 - \hat{\theta}_2$ in the elbow angle is plotted. (**Aii-Cii**) The control input was chosen to perform a swinging acrobot-like task by applying small torque only on the elbow joint. (**Cii**) The shoulder angle $\theta_1(t)$ is plotted versus the elbow angle $\theta_2(t)$ for the reference (blue) and the network (red) for the full duration in Aii-Bii. The green arrow shows the starting direction. (**D**) Reaching task. Snapshots of the configuration of the arm, reference in blue (top panels) and network in red (bottom panels) subject to torques in the directions shown by the circular arrows. After 0.6 s, the tip of the forearm reaches the cyan target. Gravity acts downwards in the direction of the arrow. (**E**) Acrobot-inspired swinging task (visualization of panels of Aii-Cii). Analogous to D, except that the torque is applied only at the elbow. To reach the target, the arm swings forward, back, and forward again.

DOI: https://doi.org/10.7554/eLife.28295.012

amplitudes drawn uniformly from an interval, changing on two time scales, 50 ms and 4 s (see Methods).

During the four seconds before learning started, we blocked error feedback. Because of zero error feedback and our initialization with zero feedforward and recurrent weights, the output $\hat{x}$ decoded from the network of spiking neurons remained constant at zero while the reference system performed the desired oscillations. Once the error feedback with large gain ($k = 10$) was turned on, the feedback forced the network to roughly follow the reference. Thus, with feedback, the error dropped to a very low value, immediately after the start of learning (*Figure 2B,C*). During learning, the error dropped even further over time (*Figure 2D*). After having stopped learning at 5000 s ($\sim 2$ hr), we found the weight distribution to be uni-modal with a few very large weights (*Figure 2G*). In the open-loop testing phase without error feedback, a sharp square pulse as initial input on different 4 s long pedestal values caused the network to track the reference as shown in *Figure 2Aii–Cii* panels. For some values of the constant pedestal input, the phase of the output of the recurrent network differed from that of the reference (*Figure 2Bii*), but the shape of the non-linear oscillation was well predicted as indicated by the similarity of the trajectories in state space (*Figure 2Cii*).

The spiking pattern of neurons of the recurrent network changed as a function of time, with inter-spike intervals of individual neurons correlated with the output, and varying over time (*Figure 2H,I*).

The distributions of firing rates averaged over a 0.25 s period with fairly constant output, and over a 16 s period with time-varying output, were long-tailed, with the mean across neurons maintained at approximately 12–13 Hz (*Figure 2E,F*). The distribution averaged over 16 s had a smaller number of neurons firing at very low and very high rates compared to the distribution over 0.25 s, consistent with the expectation that the identity of low-rate and high-rate neurons changed over time for time-varying output (*Figure 2E,F*). We repeated this example experiment ('van der Pol oscillator') with a network of equal size but with neurons that had higher firing rates, so that some neurons could reach a maximal rate of 400 Hz (*Figure 1—figure supplement 1*). The reference was approximated better and learning time was shorter with higher rates (*Figure 2—figure supplement 1* – 10,000 s with constant learning rate) compared to the low rates here (*Figure 2* – 5,000 s with 20 times the learning rate after 1,000 s). Hence, for all further simulations, we set neuronal parameters to enable peak firing rates up to 400 Hz (*Figure 1—figure supplement 1B*).

We also asked whether merely the distribution of the learned weights in the recurrent layer was sufficient to perform the task, or whether the specific learned weight matrix was required. This question was inspired from reservoir computing (*Jaeger, 2001*; *Maass et al., 2002*; *Legenstein et al., 2003*; *Maass and Markram, 2004*; *Jaeger and Haas, 2004*; *Joshi and Maass, 2005*; *Legenstein and Maass, 2007*), where the recurrent weights are random, and only the readout weights are learned. To answer this question, we implemented a perceptron learning rule on the readout weights initialized at zero, with the learned network's output as the target, after setting the feedforward and/or recurrent weights to either the learned weights as is or after shuffling them. The readout weights could be approximately learned only for the network having the learned weights and not the shuffled ones (*Figure 2—figure supplement 3*), supporting the view that the network does not behave like a reservoir (Methods).

### Chaotic Lorenz system

Our FOLLOW scheme also enabled a network with 5000 neurons each in the command representation layer and recurrent network, to learn the 3-dimensional non-linear chaotic Lorenz system (*Figure 3*). We considered a paradigm where the command input remained zero so that the network had to learn the autonomous dynamics characterized in chaos theory as a 'strange attractor' (*Lorenz, 1963*). During the testing phase without error feedback minor differences led to different trajectories of the network and the reference which show up as large fluctuations of $\epsilon_3(t)$ (*Figure 3A–C*). Such a behaviour is to be expected for a chaotic system where small changes in initial condition can lead to large changes in the trajectory. Importantly, however, the activity of the spiking network exhibits qualitatively the same underlying strange attractor dynamics, as seen from the butterfly shape (*Lorenz, 1963*) of the attractor in configuration space, and the tent map (*Lorenz, 1963*) of successive maxima versus the previous maxima (*Figure 3D,E*). The tent map generated from our network dynamics (*Figure 3E*) has lower values for the larger maxima compared to the reference tent map. However, very large outliers like those seen in a network trained by FORCE (*Thalmeier et al., 2016*) are absent. Since we expected that the observed differences are due to the filtering of the reference by an exponentially-decaying filter, we repeated learning without filtering the Lorenz reference signal (*Figure 3—figure supplement 1*), and found that the mismatch for large maxima reduced, but a doubling appeared in the tent map (*Figure 3—figure supplement 1E*) which had been almost imperceptible with filtering (cf. *Figure 3E*).

## FOLLOW enables learning a two-link planar arm model under gravity

To turn to a task closer to real life, we next wondered if a spiking network can also learn the dynamics of a two-link arm via the FOLLOW scheme. We used a two-link arm model adapted from (*Li, 2006*) as our reference. The two links in the model correspond to the upper and fore arm, with the elbow joint in between and the shoulder joint at the top. The arm moved in the vertical plane under gravity, while torques were applied directly at the two joints, so as to coarsely mimic the action of muscles. To avoid full rotations, the two joints were constrained to vary in the range from $-90°$ to $+90°$ where the resting state is at $0°$ (see Methods).

The dynamical system representing the arm is four-dimensional with the state variables being the two joint angles and two angular velocities. The network must integrate the torques to obtain the angular velocities which in turn must be integrated for the angles. Learning these dynamics is

difficult due to these sequential integrations involving non-linear functions of the state variables and the input. Still, our feedforward and recurrent network architecture (*Figure 1B*) with 5000 neurons in each layer was able to approximate these dynamics.

Similar to the previous examples, random input torque with amplitudes of short and long pulses changing each 50 ms and 1 s, respectively, was provided to each joint during the learning phase. The input was linearly interpolated between consecutive values drawn every 50 ms. In the closed loop scenario with error feedback, the trajectory converged rapidly to the target trajectory (*Figure 4*). We found that the FOLLOW scheme learned to reproduce the arm dynamics even without error feedback for a few seconds during the test phase (*Figure 4* and *Video 1* and *Video 2*), which corresponds to the time horizon needed for the planning of short arm movements.

To assess the generalization capacity of the network, we fixed the parameters post learning, and tested the network in the open-loop setting on a reaching task and an acrobot-inspired swinging task (*Sutton, 1996*). In the reaching task, torque was provided to both joints to enable the arm-tip to reach beyond a specific $(x, y)$ position from rest. The arm dynamics of the reference model and the network are illustrated in *Figure 4D* and animated in *Video 1*. We also tested the learned network model of the 2-link arm on an acrobot-like task that is a gymnast swinging on a high-bar (*Sutton, 1996*), with the shoulder joint analogous to the hands on the bar, and the elbow joint to the hips. The gymnast can only apply small torques at the hip and none at the hands, and must reach beyond a specified $(x, y)$ position by swinging. Thus, during the test, we provided input only at the elbow joint, with a time course that could make the reference reach beyond the target $(x, y)$ position from rest by swinging. The control input and the dynamics (*Figure 4A–C* right panels, *Figure 4E* and *Video 2*) show that the network can perform the task in open-loop condition suggesting that it has learned the inertial properties of the arm model, necessary for this simplified acrobot task.

## Feedback in the FOLLOW scheme entrains spike timings

In Methods, we show that the FOLLOW learning scheme is Lyapunov stable and that the error tends to zero under certain reasonable assumptions and approximations. Two important assumptions of the proof are that the weights remain bounded and that the desired dynamics are realizable by the network architecture, that is there exist feedforward and recurrent weights that enable the network to mimic the reference dynamics perfectly. However, in practice the realizability is limited by at least two constraints. First, even in networks of $N$ rate neurons with non-linear tuning curves, the non-linear function $\vec{h}$ of the reference system in *Equation (1)* can in general only be approximated with a finite error (*Funahashi, 1989*; *Girosi and Poggio, 1990*; *Hornik et al., 1989*; *Sanner and Slotine, 1992*; *Eliasmith and Anderson, 2004*) which can be interpreted as a form of frozen noise, that is even with the best possible setting of the weights, the network predicts, for most values of the state variables, a next state which is slightly different than the one generated by the reference differential equation. Second, since we work with spiking neurons, we expect on top of this frozen noise the effect of shot noise caused by pseudo-random spiking. Both noise sources may potentially cause drift of the weights (*Narendra and Annaswamy, 1989*; *Ioannou and Sun, 2012*) which in turn can make the weights grow beyond any reasonable bound. Ameliorative techniques from adaptive control are discussed in Appendix 1. In our simulations, we did not find any effect of drift of weights on the error during a learning time up to 100,000 s (*Figure 5A*), 10 times longer than that required

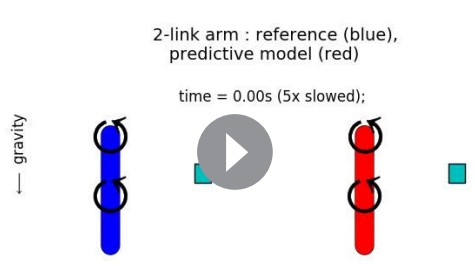

**Video 1.** Reaching by the reference arm is predicted by the network. After training the network as a forward model of the two-link arm under gravity as in *Figure 4*, we tested the network without feedback on a reaching task. Command input was provided to both joints of the two-link reference arm so that the tip reached the cyan square. The same command input was also provided to the network without error feedback. The state (blue, left) of the reference arm and the state predicted (red, right) by the learned network without error feedback are animated as a function of time. The directions of the circular arrows indicate the directions of the command torques at the joints. The animation is slowed 5× compared to real life.

DOI: https://doi.org/10.7554/eLife.28295.014

for learning this example (*Figure 2—figure supplement 1*).

To highlight the difference between a realizable reference system and non-linear differential equations as a reference system, we used, in an additional simulation experiment, a spiking network with fixed weights as the reference. More precisely, instead of using directly the differential equations of the van der Pol oscillator as a reference, we now used as a reference a spiking approximation of the van der Pol oscillator, that is the spiking network that was the final result after 10,000 s (~3 hr) of FOLLOW learning in *Figure 2—figure supplement 1*. For both the spiking reference network and the to-be-trained learning network we used the same architecture, the same number of neurons, and the same neuronal parameters as in *Figure 2—figure supplement 1* for the learning of the van der Pol oscillator. The readout and feedback weights of the learning network also had the same parameters as those of the spiking reference network, but the feedforward and recurrent weights of the learning network were initialized to zero and updated, during the learning phase, with the FOLLOW rule. We ran FOLLOW learning against the reference network for 100,000 s (~28 hr) (*Figure 5*). With the realizable network as reference, learning was more rapid than with the original van der Pol oscillator as reference (*Figure 5A*).

We emphasize that, analogous to the earlier simulations, the feedback error $\epsilon_\alpha$ was low-dimensional and calculated from the decoded outputs. Nevertheless, the low-dimensional error feedback was able to entrain the network spike times to the reference spike times (*Figure 5C*). In particular, a few neurons learned to fire only two or three spikes at very precise moments in time. For example, after learning, the spikes of neuron $i = 9$ in the learning network were tightly aligned with the spike times of the neuron with the same index $i$ in the spiking reference network. Similarly, neuron $i = 8$ that was inactive at the beginning of learning was found to be active, and aligned with the spikes of the reference network, after 100,000 s (~28 hr) of learning. The spike trains were entrained by the low-dimensional feedback. With the feedback off, even the low-dimensional output, and hence the spike trains, diverged from the reference. It will be interesting to explore if this entrainment by low-dimensional feedback via an auto-encoder loop can be useful in supervised spike train learning (*Gütig and Sompolinsky, 2006*; *Pfister et al., 2006*; *Florian, 2012*; *Mohemmed et al., 2012*; *Gütig, 2014*; *Memmesheimer et al., 2014*; *Gardner and Grüning, 2016*).

Our results with the spiking reference network suggest that the error is reduced to a value close to zero for a realizable or closely-approximated system (*Figure 5A*) as shown in Methods, analogous to proofs in adaptive control (*Ioannou and Sun, 2012*; *Narendra and Annaswamy, 1989*). Moreover, network weights became very similar, though not completely identical, to the weights of the realizable reference network (*Figure 5B*), which suggests that the theorem for convergence of parameters from adaptive control should carry over to our learning scheme.

## Learning is robust to sparse connectivity, noisy error or reference, and noisy decoding weights, but not to delays

So far, our spiking networks had all-to-all connectivity. We next tested whether sparse connectivity (*Markram et al., 2015*; *Brown and Hestrin, 2009*) of the feedforward and recurrent connections was sufficient for learning low-dimensional dynamics. We ran the van der Pol oscillator learning protocol with the connectivity varying from 0.1 (10 percent connectivity) to 1 (full connectivity). Connections that were absent after the sparse initialization could not appear during learning, while the existing sparse connections were allowed to evolve according to FOLLOW learning. As shown in *Figure 6A*, we found that learning was slower with sparser connectivity; but with twice the learning time, a sparse network with about 25% connectivity reached similar performance as the fully connected network with standard learning time.

We added Gaussian white noise to each component of the error, which is equivalent to adding it to each component of the reference, and ran the van der Pol oscillator learning protocol for 10,000 s for different standard deviations of the noise (*Figure 6B*). The learning was robust to noise with standard deviation up to around $0.001$, which must be compared with the error amplitude of the order of $0.1$ at the start of learning, and orders of magnitude lower later.

The readout weights have been pre-learned until now, so that, in the absence of recurrent connections, error feedback weights and decoding weights formed an auto-encoder. We sought to relax this requirement. Simulations showed that with completely random readout weights, the system did not learn to reproduce the target dynamical system. However, if the readout weights had some overlap with the auto-encoder, learning was still possible (*Figure 6C*). If for a feedback error $\vec{\epsilon}$, the

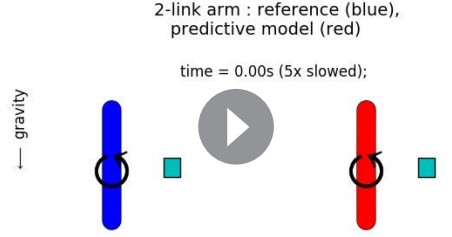

2-link arm : reference (blue),
predictive model (red)

time = 0.00s (5x slowed);

← gravity

**Video 2.** Acrobot-like swinging by the reference arm is predicted by the network. After training the network as a forward model of the two-link arm under gravity as in *Figure 4*, we tested the network without feedback on a swinging task analogous to an acrobot. Command input was provided to the elbow joint of the two-link reference arm so that the tip reached the cyan square by swinging. The same command input was also provided to the network without error feedback. The state (blue, left) of the reference arm and the state predicted (red, right) by the learned network without error feedback are animated as a function of time. The directions of the circular arrows indicate the directions of the command torques at the joints. The animation is slowed 5× compared to real life.
DOI: https://doi.org/10.7554/eLife.28295.015

error encoding followed by output decoding yields $k(1 + \xi)\vec{\epsilon} + \vec{n}(\vec{\epsilon})$, where $\vec{n}$ is a vector of arbitrary functions not having linear terms and small in magnitude compared to the first term, and $\xi$ is sufficiently greater than $-1$ so that the effective gain $k(1 + \xi)$ remains large enough, then the term that is linear in error can still drive the output close to the desired one (see Methods).

To check this intuition in simulations, we incorporated multiplicative noise on the decoders by multiplying each decoding weight of the auto-encoder by one plus $\gamma$, where for each weight $\gamma$ was drawn independently from a uniform distribution between $-\chi + \xi$ and $\chi + \xi$. We found that the system was still able to learn the van der Pol oscillator up to $\chi \sim 5$ and $\xi = 0$, or $\chi = 2$ and $\xi$ variable (*Figure 6B,C*). Negative values of $\xi$ result in a lower overlap with the auto-encoder leading to the asymmetry seen in *Figure 6C*. Thus, the FOLLOW learning scheme is robust to multiplicative noise on the decoding weights. Alternative approaches for other noise models are discussed in Appendix 1.

We also asked if the network could handle sensory feedback delays in the reference signal. Due to the strong limit cycle attractor of the van der Pol oscillator, the effect of delay is less transparent than for the linear decaying oscillator (*Figure 2—figure supplement 2*), so we decided to focus on the latter. For the linear decaying oscillator, we found that learning degraded rapidly with a few milliseconds of delay in the reference, that is if $\vec{x}(t - \Delta)$ was provided as reference instead of $\vec{x}(t)$ (*Figure 6E–F*). We compensated for the sensory feedback delay by delaying the motor command input by identical $\Delta$ (*Figure 6G*), which is equivalent to time-translating the complete learning protocol, to which the learning is invariant, and thus the network would learn for arbitrary delay (*Figure 6H*). In the Discussion, we suggest how a forward model learned with a compensatory delay (*Figure 6G*) could be used in control mode to compensate for sensory feedback delays.

## Discussion

The FOLLOW learning scheme enables a spiking neural network to function as a forward predictive model that mimics a non-linear dynamical system activated by one or several time-varying inputs. The learning rule is supervised, local, and comes with a proof of stability.

It is supervised because the FOLLOW learning scheme uses error feedback where the error is defined as the difference between predicted output and the actual observed output. Error feedback forces the output of the system to mimic the reference, an effect that is widely used in adaptive control theory (*Narendra and Annaswamy, 1989*; *Ioannou and Sun, 2012*).

The learning rule is local in the sense that it combines information about presynaptic spike arrival with an abstract quantity that we imagine to be available in the postsynaptic neuron. In contrast to standard Hebbian learning, the variable representing this postsynaptic quantity is not the postsynaptic firing rate, spike time, or postsynaptic membrane potential, but the error current projected by feedback connections onto the postsynaptic neuron, similar in spirit to modern biological implementations of approximated backpropagation (*Roelfsema and van Ooyen, 2005*; *Lillicrap et al., 2016*) or local versions of FORCE (*Sussillo and Abbott, 2009*) learning rules. We emphasize that the postsynaptic quantity is different from the postsynaptic membrane potential or the total postsynaptic current which would also include input from feedforward and recurrent connections.

A possible implementation in a spatially extended neuron would be to imagine that the postsynaptic error current $I_i^{\epsilon}$ arrives in the apical dendrite where it stimulates messenger molecules that

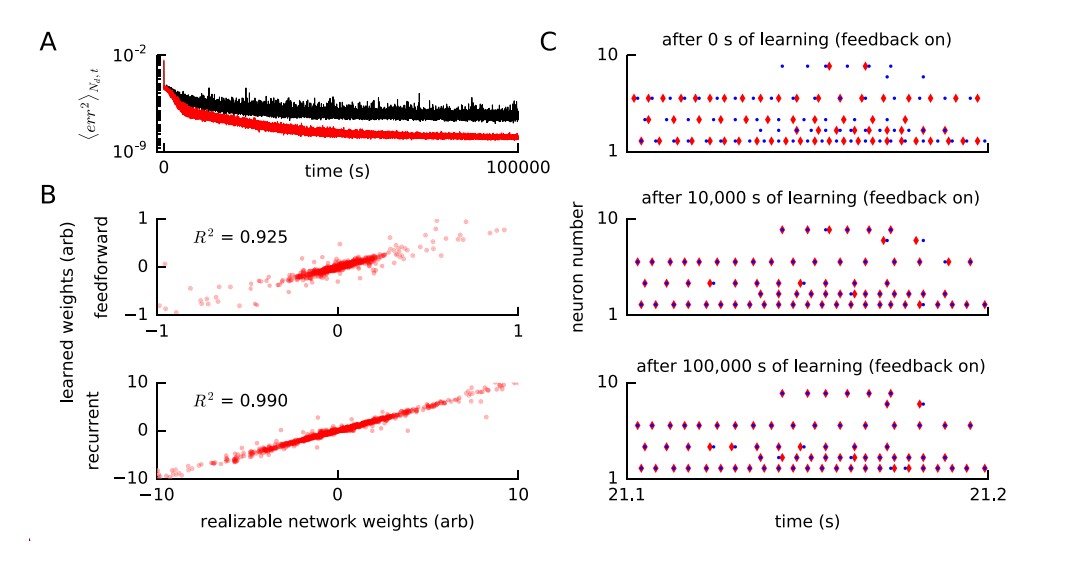

**Figure 5.** Convergence of error, weights and spike times for a realizable reference network. (**A**) We ran our FOLLOW scheme on a network for learning one of two different implementations of the reference van der Pol oscillator: (1) differential equations, versus (2) a network realized using FOLLOW learning for 10,000 s (~3 hr). We plot the evolution of the mean squared error, mean over number of dimensions $N_d$ and over 4 s time blocks, from the start to 100,000 s of learning, with the weights starting from zero. Mean squared error for the differential equations reference (1) is shown in black, while that for the realizable network reference (2) is in red. (**B**) The feedforward weights (top panel) and the recurrent weights (bottom panel) at the end of 100,000 s (~28 hr) of learning, are plotted versus the corresponding weights of the realizable target network. The coefficient of determination, that is the $R^2$ value of the fit to the identity line ($y = x$), is also displayed for each panel. A value of $R^2 = 1$ denotes perfect equality of weights to those of the realizable network. Some weights fall outside the plot limits. (**C**) After 0 s, 10,000 s (~3 hr), and 100,000 s (~28 hr) of the learning protocol against the realizable network as reference, we show spike trains of a few neurons in the recurrent network (red) and the reference network (blue) in the top, middle and bottom panels respectively, from test simulations while providing the same control input and keeping error feedback on. With error feedback off, the low-dimensional output diverged slightly from the reference, hence the spike trains did too (not shown).

DOI: https://doi.org/10.7554/eLife.28295.013

quickly diffuse or are actively transported into the soma and basal dendrites where synapses from feedfoward and feedback input could be located, as depicted in *Figure 7A*. Consistent with the picture of a messenger molecule, we low-pass filtered the error current with an exponential filter $\kappa^\epsilon$ of time constant 80 ms or 200 ms, much longer than the synaptic time constant of 20 ms of the filter $\kappa$. Simultaneously, filtered information about presynaptic spike arrival $S_j * \kappa$ is available at each synapse, possibly in the form of glutamate bound to the postsynaptic receptor or by calcium triggered signalling chains localized in the postsynaptic spines. Thus the combination of effects caused by presynaptic spike arrival and error information available in the postsynaptic cell drives weight changes, in loose analogy to standard Hebbian learning.

The separation of the error current from the currents at feedforward or recurrent synapses could be spatial (such as suggested in *Figure 7A*) or chemical if the error current projects onto synapses that trigger a signalling cascade that is different from that at other synapses. Importantly, whether it is a spatial or chemical separation, the signals triggered by the error currents need to be available throughout the postsynaptic neuron. This leads us to a prediction regarding synaptic plasticity that, say in cortical pyramidal neurons, the plasticity of synapses that are driven by pre-synaptic input in the basal dendrites, should be modulated by currents injected in the apical dendrite or on stimulation of feedback connections.

The learning scheme is provenly stable with errors converging asymptotically to zero under a few reasonable assumptions (Methods). The first assumption is that error encoding feedback weights and output decoding readout weights form an auto-encoder. This requirement can be met if, at an early developmental stage, either both sets of weights are learned using say mirrored STDP (*Burbank, 2015*), or the output readout weights are learned, starting with random encoding weights, via a biological perceptron-like learning rule (*D'Souza et al., 2010*; *Urbanczik and Senn, 2014*). A pre-learned auto-encoder in a high-gain negative feedback loop is in fact a specific prediction of our

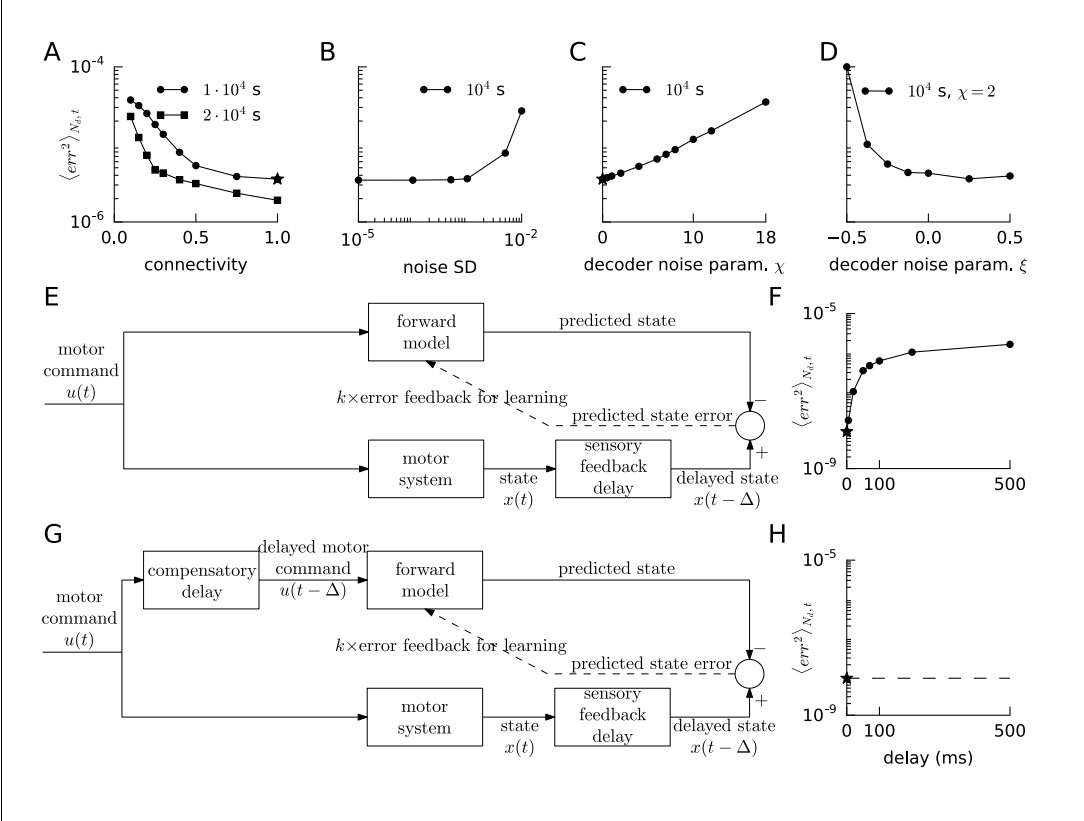

**Figure 6.** Robustness of FOLLOW learning. We ran the van der Pol oscillator (**A–D**) or the linear decaying oscillator (**F,H**) learning protocol for 10,000 s for different parameter values and measured the mean squared error, over the last 400 s before the end of learning, mean over number of dimensions $N_d$ and time. (**A**) We evolved only a fraction of the feedforward and recurrent connections, randomly chosen as per a specific connectivity, according to FOLLOW learning, while keeping the rest zero. The round dots show mean squared errors for different connectivities after a 10,000 s learning protocol (default connectivity = 1 is starred); while the square dots show the same after a 20,000 s protocol. (**B**) Mean squared error after 10,000 s of learning versus the standard deviation of noise added to each component of the error, or equivalently to each component of the reference, is plotted. (**C**) We multiplied the original decoding weights (that form an auto-encoder with the error encoders) by a random factor (1 + uniform($-\chi, \chi$)) drawn for each weight. The mean squared error at the end of a 10,000 s learning protocol for increasing values of $\chi$ is plotted (default $\chi = 0$ is starred). (**D**) We multiplied the original decoding weights by a random factor (1 + uniform($-\chi + \xi, \chi + \xi$)), fixing $\chi = 2$, drawn independently for each weight. The mean squared error at the end of a 10,000 s learning protocol, for a few values of $\xi$ on either side of zero, is plotted. (**E,G**) Architectures for learning the forward model when the reference $x(t)$ is available after a sensory feedback delay $\Delta$ for computing the error feedback. The forward model may be trained without a compensatory delay in the motor command path (**E**) or with it (**G**). (**F,H**) Mean squared error after 10,000 s of learning the linear decaying oscillator is plotted (default values are starred) versus the sensory feedback delay $\Delta$ in the reference, for the architectures without and with compensatory delay, in F and H respectively.

DOI: https://doi.org/10.7554/eLife.28295.016

learning scheme, to be tested in systems-level experiments. The second assumption is that the reference dynamics $f(\vec{x})$ is realizable. This requirement can be approximately met by having a recurrent network with a large number $N$ of neurons with different parameters (*Eliasmith and Anderson, 2004*). The third assumption is that the state variables $\vec{x}(t)$ are observable. While currently we calculate the feedback error directly from the state variables as a difference between reference and predicted state, we could soften this condition and calculate the difference in a higher-dimensional space with variables $\vec{y}(t)$ as long as $\vec{y} = K(\vec{x})$ is an invertible function of $\vec{x}(t)$ (Appendix 1). The fourth assumption is that the system dynamics be slower than synaptic dynamics. Indeed, typical reaching movements extend over hundreds of milliseconds or a few seconds whereas neuronal spike transmission delays and synaptic time constants can be as short as a few milliseconds. In our simulations, neuronal and synaptic time constants were set to 20 ms, yet the network dynamics evolved on the time scale of hundreds of milliseconds or a few seconds, even in the open-loop condition when error feedback was switched off (*Figures 2* and *4*). The fifth assumption is that weights stay bounded.

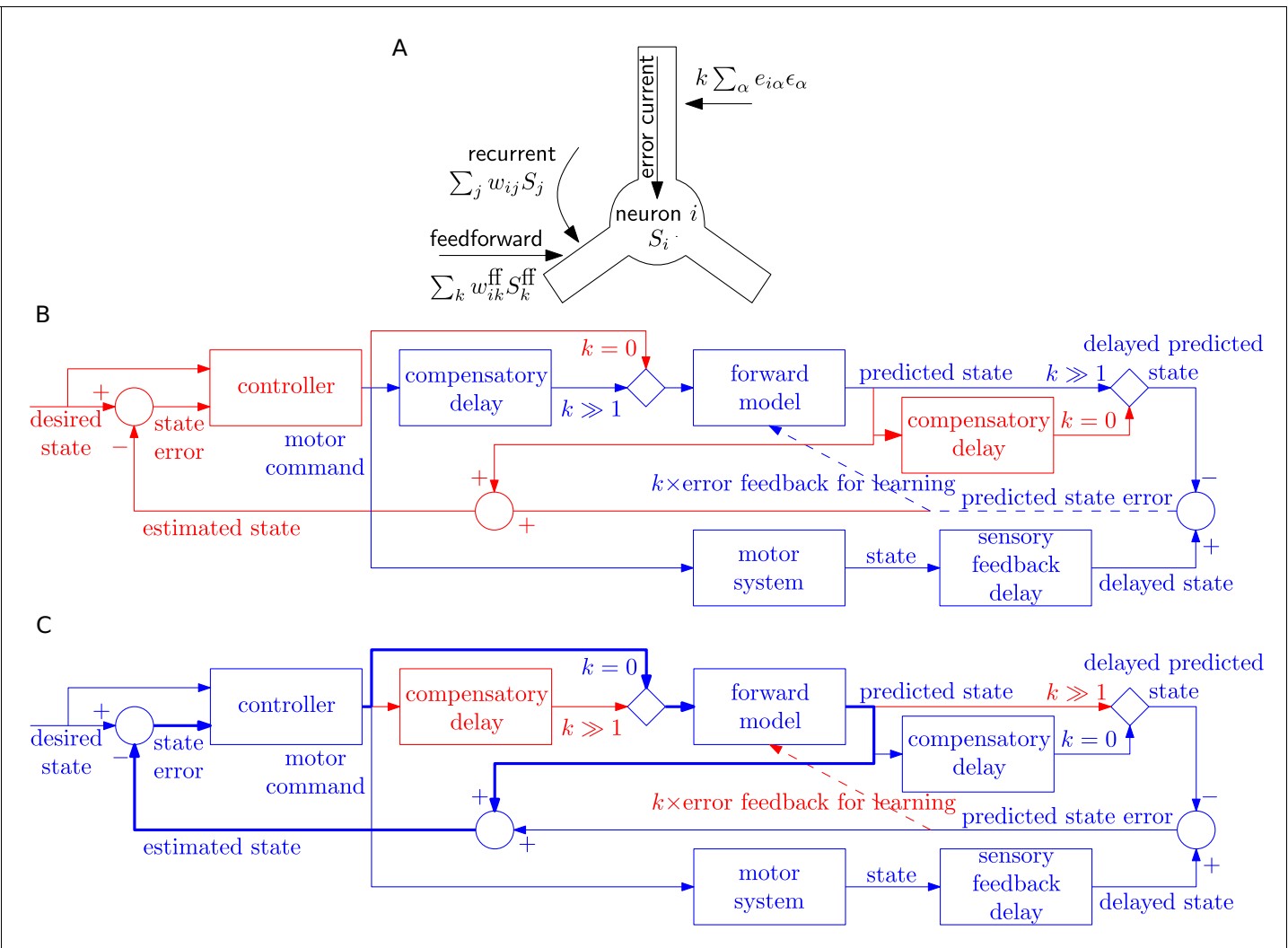

**Figure 7.** Possible implementation of learning rule, and delay compensation using forward model. (**A**) A cartoon depiction of feedforward, recurrent and error currents entering a neuron $i$ in the recurrent network. The error current enters the apical dendrite and triggers an intra-cellular chemical cascade generating a signal that is available at the feedforward and recurrent synapses in the soma and basal dendrites, for weight updates. The error current must trigger a cascade isolated from the other currents, here achieved by spatial separation. (**B-C**) An architecture based on the Smith predictor, that can switch between learning the forward model (B), versus using the forward model for motor control (C, adapted from (***Wolpert and Miall, 1996***)), to compensate for the delay in sensory feedback. Active pathways are in blue and inactive ones are in red. (**B**) The learning architecture (blue) is identical to ***Figure 6G***, but embedded within a larger control loop (red). During learning, when error feedback gain $k \gg 1$, the motor command is fed in with a compensatory delay identical to the sensory feedback delay. Thus motor command and reference state are equally delayed, hence temporally matched, and the forward model learns to produce the motor system output for given input. (**C**) Once the forward model is learned, the system switches to motor control mode (feedback gain $k = 0$). In this mode, the forward model receives the present motor command and predicts the current state of the motor system, for rapid feedback to the controller (via loop indicated by thick lines), even before the delayed sensory feedback arrives. Of course the delayed sensory feedback can be further taken into account by the controller, by comparing it with the delayed output of the forward model, to better estimate the true state. Thus the forward model learned as in B provides a prediction of the state, even before feedback is received, acting to compensate for sensory feedback delays in motor control.

DOI: https://doi.org/10.7554/eLife.28295.017

Indeed, in biology, synaptic weights should not grow indefinitely. Algorithmically, a weight decay term in the learning rule can suppress the growth of large weights (see also Appendix 1), though we did not need to implement a weight decay term in our simulations.

One of the postulated uses of the forward predictive model is to compensate for delay in the sensory feedback during motor control (***Wolpert and Miall, 1996***; ***Wolpert et al., 1995***) using the Smith predictor configuration (***Smith, 1957***). We speculate that the switch from the closed-loop

learning of forward model with feedback gain $k \gg 1$ to open-loop motor prediction $k = 0$ could also be used to switch delay lines: the system can have either a delay before the forward model as required for learning (*Figure 7B*), or after the forward model as required for the Smith predictor (*Figure 7C*). We envisage that FOLLOW learning of the forward model occurs in closed loop mode ($k \gg 1$) with a delay in the motor command path, as outlined earlier in *Figure 6G* and now embedded in the Smith predictor architecture in *Figure 7B*. After learning, the network is switched to motor control mode, with the forward predictive model in open loop ($k = 0$), implementing the Smith predictor (*Figure 7C*). In this motor control mode, the motor command is fed with zero delay to the forward model. This enables to rapidly feed the estimated state back to the motor controller so as to take corrective actions, even before sensory feedback arrives. In parallel, available sensory feedback is compared with a copy of the forward model that has passed through a compensatory delay after the forward model (*Figure 7C*).

Simulations with the FOLLOW learning scheme have demonstrated that strongly non-linear dynamics can be learned in a recurrent spiking neural network using a local online learning rule that does not require rapid weight changes. Previous work has mainly focused on a limited subset of these aspects. For example, Eliasmith and colleagues used a local learning rule derived from stochastic gradient descent, in a network structure comprising heterogeneous spiking neurons with error feedback (*MacNeil and Eliasmith, 2011*), but did not demonstrate learning non-linear dynamics (Appendix 1). Denève and colleagues used error feedback in a homogeneous spiking network with a rule similar to ours, for linear dynamics only (*Bourdoukan and Denève, 2015*), and while this article was in review, also for non-linear dynamics (*Alemi et al., 2017*), but their network requires instantaneous lateral interactions and in the latter case, also non-linear dendrites.

Reservoir computing models exploit recurrent networks of non-linear units in an activity regime close to chaos where temporal dynamics is rich (*Jaeger, 2001*; *Maass et al., 2002*; *Legenstein et al., 2003*; *Maass and Markram, 2004*; *Jaeger and Haas, 2004*; *Joshi and Maass, 2005*; *Legenstein and Maass, 2007*). While typical applications of reservoir computing are concerned with tasks involving a small set of desired output trajectories (such as switches or oscillators), our FOLLOW learning enables a recurrent network with a single set of parameters to mimic a dynamical system over a broad range of time-dependent inputs with a large family of different trajectories in the output.

Whereas initial versions of reservoir computing focused on learning the readout weights, applications of FORCE learning to recurrent networks of rate units made it possible to also learn the recurrent weights (*Sussillo and Abbott, 2009, 2012*). However, in the case of a multi-dimensional target, multi-dimensional errors were typically fed to distinct parts of the network, as opposed to the distributed encoding used in our network. Moreover, the time scale of synaptic plasticity in FORCE learning is faster than the time scale of the dynamical system which is unlikely to be consistent with biology. Modern applications of FORCE learning to spiking networks (*DePasquale et al., 2016*; *Thalmeier et al., 2016*; *Nicola and Clopath, 2016*) inherit these issues.

Adaptive control of non-linear systems using continuous rate neurons (*Sanner and Slotine, 1992*; *Weiping Li et al., 1987*; *Slotine and Coetsee, 1986*) or spiking neurons (*DeWolf et al., 2016*) has primarily focused on learning parameters in adaptive feedback paths, rather than learning weights in a recurrent network, using learning rules involving quantities that do not appear in the pre- or post-synaptic neurons, making them difficult to interpret as local to synapses. Recurrent networks of rate units have occasionally been used for control (*Zerkaoui et al., 2009*), but trained either via real-time recurrent learning or the extended Kalman filter which are non-local in space, or via backpropagation through time which is offline (*Pearlmutter, 1995*). Recent studies have used neural network techniques to train inverse models by motor babbling, to describe behavioral data in humans (*Berniker and Kording, 2015*) and song birds (*Hanuschkin et al., 2013*), albeit with abstract networks. Optimal control methods (*Hennequin et al., 2014*) or stochastic gradient descent (*Song et al., 2016*) have also been applied in recurrent networks of neurons, but with limited biological plausibility of the published learning rules. As an alternative to supervised schemes, biologically plausible forms of reward-modulated Hebbian rules on the output weights of a reservoir have been used to learn periodic pattern generation and abstract computations (*Hoerzer et al., 2014*; *Legenstein et al., 2010*), but how such modulated Hebbian rules could be used in predicting non-linear dynamics given time-dependent control input remains open.

Additional features of the FOLLOW learning scheme are that it does not require full connectivity but also works with biologically more plausible sparse connectivity; and it is robust to multiplicative noise in the output decoders, analogous to recent results on approximate error backpropagation in artificial neural networks (*Lillicrap et al., 2016*). Since the low-dimensional output and all neural currents are spatially averaged over a large number of synaptically-filtered spike trains, neurons in the FOLLOW network do not necessarily need to fire at rates higher than the inverse of the synaptic time scale. In conclusion, we used a network of heterogeneous neurons as in the Neural Engineering Framework (*Eliasmith and Anderson, 2004*), employed a pre-learned auto-encoder to enable negative feedback of error as in adaptive control theory (*Morse, 1980*; *Narendra et al., 1980*; *Slotine and Coetsee, 1986*; *Weiping Li et al., 1987*; *Narendra and Annaswamy, 1989*; *Sastry and Bodson, 1989*; *Ioannou and Sun, 2012*), and derived and demonstrated a local and online learning rule for recurrent connections that learn to reproduce non-linear dynamics.

Our present implementation of the FOLLOW learning scheme in spiking neurons violates Dale's law because synapses originating from the same presynaptic neuron can have positive or negative weights, but in a different context extensions incorporating Dale's law have been suggested (*Parisien et al., 2008*). Neurons in cortical networks are also seen to maintain a balance of excitatory and inhibitory incoming currents (*Denève and Machens, 2016*). It would be interesting to investigate a more biologically plausible extension of FOLLOW learning that maintains Dale's law; works in the regime of excitatory-inhibitory balance, possibly using inhibitory plasticity (*Vogels et al., 2011*); pre-learns the auto-encoder, potentially via mirrored STDP (*Burbank, 2015*); and possibly implements spatial separation between different compartments (*Urbanczik and Senn, 2014*). It would also be interesting for future work to see whether our model of an arm trained on motor babbling with FOLLOW, can explain aspects of human behavior in reaching tasks involving force fields (*Shadmehr and Mussa-Ivaldi, 1994*), uncertainty (*Körding and Wolpert, 2004*; *Wei and Körding, 2010*) or noise (*Burge et al., 2008*). Further directions worth pursuing include learning multiple different dynamical transforms within one recurrent network, without interference; hierarchical learning with stacked recurrent layers; and learning the inverse model of motor control so as to generate the control input given a desired state trajectory.

## Methods

### Simulation software

All simulation scripts were written in python (https://www.python.org/) for the Nengo simulator (*Stewart et al., 2009*) (http://www.nengo.ca/, version 2.4.0) with minor custom modifications to support sparse weights. We ran the model using the Nengo GPU back-end (https://github.com/nengo/nengo_ocl) for speed. The script for plotting the figures was written in python using the matplotlib module (http://matplotlib.org/). These simulation and plotting scripts are available online at https://github.com/adityagilra/FOLLOW (*Gilra, 2017*). A copy is archived at https://github.com/elifesciences-publications/FOLLOW.

### Network parameters
#### Initialization of plastic weights
The feedforward weights $w_{il}^{\text{ff}}$ from the command representation layer to the recurrent network and the recurrent weights $w_{ij}$ inside the network were initialized to zero.

#### Update of plastic weights
With the error feedback loop closed, that is with reference output $\vec{x}$ and predicted output $\hat{x}$ connected to the error node, and feedback gain $k = 10$, the FOLLOW learning rule, *Equation (10)*, was applied on the feedforward and recurrent weights, $w_{il}^{\text{ff}}$ and $w_{ij}$. The error for our learning rule was the error $\epsilon_\alpha = x_\alpha - \hat{x}_\alpha$ in the observable output $\vec{x}$, not the error in the desired function $\vec{h}(\vec{x}, \vec{u})$ (cf. *Eliasmith, 2005*; *MacNeil and Eliasmith, 2011*, Appendix 1). The observable reference state $\vec{x}$ was obtained by integrating the differential equations of the dynamical system. The synaptic time constant $\tau_s$ was 20 ms in all synapses, including those for calculating the error and for feeding the error

back to the neurons (decaying exponential $\kappa$ with time constant $\tau_s$ in *Equation (6)*). The error used for the weight update was filtered by a 200 ms decaying exponential ($\kappa^\epsilon$ in *Equation (10)*).

## Random setting of neuronal parameters and encoding weights

We used leaky integrate-and-fire neurons with a threshold $\theta = 1$ and time constant $\tau_m = 20$ ms. After each spike, the voltage was reset to zero, and the neuron entered an absolute refractory period of $\tau_r = 2$ ms. When driven by a constant input current $J$, a leaky integrate-and-fire neuron with absolute refractoriness fires at a rate $a = g(J)$ where $g$ is the gain function with value $g(J) = 0$ for $J \leq 1$ and

$$g(J) = 1 \Big/ \left( \tau_r + \tau_m \ln \frac{J}{J-1} \right), \quad \text{for } J > 1.$$
(11)

Our network was inhomogeneous in the sense that different neurons had different parameters as described below. The basic idea is that the ensemble of $N$ neurons, with different parameters, forms a rich set of basis functions in the $N_c$ or $N_d$ dimensional space of inputs or outputs, respectively. This is similar to tiling the space with radial basis functions, except that here we replace the radial functions by the gain functions of the LIF neurons (*Equation (11)*) each having different parameters (*Eliasmith and Anderson, 2004*). These parameters were chosen randomly once at the beginning of a simulation and kept fixed during the simulation.

For the command representation layer, we write the current $J$ into neuron $l$, in the case of a constant input $\vec{u}$, as

$$J_l^{\text{ff}} = \nu_l^{\text{ff}} \sum_\beta \tilde{e}_{l\beta}^{\text{ff}} u_\beta + b_l^{\text{ff}}, \quad \text{with } e_{l\beta}^{\text{ff}} \equiv \nu_l^{\text{ff}} \tilde{e}_{l\beta}^{\text{ff}},$$
(12)

where $\nu_l^{\text{ff}}$ and $b_l^{\text{ff}}$ are neuron-specific gains and biases, and $\tilde{e}_{l\beta}^{\text{ff}}$ are 'normalized' encoding weights (cf. *Equation (5)*).

These random gains, biases and 'normalized' encoding weights must be chosen so that the command representation layer adequately represents the command input $\vec{u}$, whose norm is bounded in the interval $[0, R_1]$ (*Table 1*). First, we choose the 'normalized' encoding weight vectors on a hypersphere of radius $1/R_1$, so that the scalar product between the command vector and the vector of

**Table 1.** Network and simulation parameters for example systems.

| | Linear | van der Pol | Lorenz | Arm | Non-linear feedforward |
|---|---|---|---|---|---|
| Number of neurons/layer | 2000 | 3000 | 5000 | 5000 | 2000 |
| $T_{period}$ (s) | 2 | 4 | 20 | 2 | 2 |
| Representation radius $R_1$ | 0.2 | 0.2 | 6 | 0.2 | 0.2 |
| Representation radius $R_2$ | 1 | 5[†] | 30 | 1 | 1 |
| Gains $\nu_i$ and biases $b_i$ for command representation and recurrent layers | Nengo v2.4.0 default [‡] | *Figures 1* and *2*: $\nu_i = 2$ and $b_i$ chosen uniformly from $[-2, 2)$. All other Figures: Nengo v2.4.0 default [‡] | Nengo v2.4.0 default [‡] | Nengo v2.4.0 default [‡] | Nengo v2.4.0 default[‡] |
| Learning pulse $\zeta_1$ | $R_1/6$ | $R_1/6, R_1/2$ | $R_2/10$ | $R_2/0.3$ | $R_1/0.6$ |
| Learning pedestal $\zeta_2$ | $R_2/16$ | $R_1/6, R_1/2$ | 0 | $R_2/0.3$ | $R_2/1.6$ |
| Learning rate $\eta$ | 2e-4 | 2e-4* | 2e-4 | 2e-4 | 2e-4 |
| Figures | *Figure 2—figure supplement 2* | *Figure 1, Figure 2, Figure 2—figure supplement 1, Figure 2—figure supplement 3, Figure 5, Figure 6, Figure 7* | *Figure 3, Figure 3—figure supplement 1* | *Figure 4* | *Figure 2—figure supplement 4, Figure 2—figure supplement 5* |

* 4e-3 after 1,000 s for *Figures 1* and *2*. 1e-4 for readout weights in *Figure 2—figure supplement 3*.

[†] 4.5 for *Figures 1* and *2*.

[‡] Nengo v2.4.0 sets the gains and biases indirectly, by default. The projected input at which the neuron just starts firing (i.e. $\sum_\alpha \tilde{e}_{i\alpha} x_\alpha = \tilde{J}_i^0$) is chosen uniformly from $[-1, 1)$, while the firing rate for $\sum_\alpha \tilde{e}_{i\alpha} x_\alpha = 1$ is chosen uniformly between 200 and 400 Hz. From these, $\nu_i$ and $b_i$ are computed using *Equations (11) and (13)*.

DOI: https://doi.org/10.7554/eLife.28295.018

'normalized' encoding weights, $\sum_\beta \tilde{e}^{\mathrm{ff}}_{l\beta} u_\beta$, lies in the normalized range $[-1, 1]$. Second, the distribution of the gains sets the distribution of the firing rates in a target range. Third, we see from *Equation (11)* that the neuron starts to fire at the rheobase threshold $J = 1$. The biases $b^{\mathrm{ff}}_l$ randomly shift this rheobase threshold over an interval (see *Figure 1—figure supplement 1*). For the distributions used to set the fixed random gains and biases, see *Table 1*.

Analogously, for the recurrent network, we write the current into neuron $i$, for a constant 'pseudo-input' vector $\vec{\tilde{x}}$ being represented in the network, as

$$J_i = \nu_i \sum_\alpha \tilde{e}_{i\alpha} \tilde{x}_\alpha + b_i, \quad \text{with } e_{i\alpha} \equiv \nu_i \tilde{e}_{i\alpha}, \tag{13}$$

where $\nu_i$, $b_i$ are neuron-specific gains and biases, and $\tilde{e}_{i\alpha}$ are 'normalized' encoding weights. We call $\vec{\tilde{x}}$ a 'pseudo-input' for two reasons. First, the error encoding weights $ke_{i\alpha}$ are used to feed the error $\epsilon_\alpha = (x_\alpha - \hat{x}_\alpha)$ back to neuron $i$ in the network (cf. *Equation (6)*). However, $\epsilon_\alpha = x_\alpha/(k+1)$, due to the feedback loop according to *Equation (9)*. Thus, the 'pseudo-input' $\tilde{x}_\alpha = kx_\alpha/(k+1)$ has a similar range as $\vec{x}$, whose norm lies in the interval $[0, R_2]$ (see *Table 1*). Second, the neuron also gets feedforward and recurrent input. However, the feedforward and recurrent inputs get automatically adjusted during learning (starting from zero), so their absolute values do not matter for the initialization of parameters that we discuss here. Thus, we choose the 'normalized' encoding weight vectors on a hypersphere of radius $1/R_2$. For the distributions used to set the fixed random gains and biases, see *Table 1*.

## Setting output decoding weights to form an auto-encoder with respect to error encoding weights

The linear readout weights $d_{\alpha i}$ from the recurrently connected network were pre-computed algorithmically so as to form an auto-encoder with the error encoding weights $e_{i\alpha}$ (for $k = 1$), while setting the feedforward and recurrent weights to zero ($w^{\mathrm{ff}}_{l\beta} = 0$ and $w_{ij} = 0$). To do this, we randomly selected $P$ error vectors $\vec{\epsilon}^{(p)}$, that we used as training samples for optimization, with sample index $p = 1, \ldots, P$, and having vector components $\epsilon^{(p)}_\alpha$, $\alpha = 1, \ldots, N_d$. Since the observable system is $N_d$-dimensional, we chose the training samples randomly from within an $N_d$-dimensional hypersphere of radius $R_2$. We applied each of the error vectors statically as input for the error feedback connections and calculated the activity

$$a^{(p)}_i \equiv a_i\left(\vec{\epsilon}^{(p)}\right) = g\left(\sum_\alpha e_{i\alpha} \epsilon^{(p)}_\alpha + b_i\right), \tag{14}$$

of neuron $i$ for error vector $\vec{\epsilon}^{(p)}$ using the static rate *Equation (11)*. The decoders $d_{\alpha i}$ acting on these activities should yield back the encoded points thus forming an auto-encoder. A squared-error loss function $\mathcal{L}$, with L2 regularization of the decoders,

$$\mathcal{L} = \frac{1}{2} \sum_p^P \sum_\alpha^{N_d} \left(\sum_i^N d_{\alpha i} a^{(p)}_i - \epsilon^{(p)}_\alpha\right)^2 + \frac{1}{2}\lambda \sum_\alpha^{N_d} \sum_i^N d^2_{\alpha i}, \tag{15}$$

setting $\lambda = P\left(0.1 \max\left(\left\{a^{(p)}_i\right\}\right)\right)^2$ with number of samples $P = N$, was used for this linear regression (default in Nengo v2.4.0) (*Eliasmith and Anderson, 2004*; *Stewart et al., 2009*). Biologically plausible learning rules exist for auto-encoders, either by training both encoding and decoding weights (*Burbank, 2015*), or by training decoding weights given random encoding weights (*D'Souza et al., 2010*; *Urbanczik and Senn, 2014*), but we simply calculated and set the decoding weights as if they had already been learned.

## Compressive and expansive auto-encoder

Classical three-layer (input-hidden-output-layer) auto-encoders come in two different flavours, viz. compressive or expansive, which have the dimensionality of the hidden layer smaller or larger respectively, than that of the input and output layers. Instead of a three-layer feedfoward network, our auto-encoder forms a loop from the neurons in the recurrent network via readout weights to the

output and from there via error-encoding weights to the input. Since the auto-encoder is in the loop, we expect that it works both as a compressive one (from neurons in the recurrent network over the low-dimensional output back to the neurons) and as an expansive one (from the output through the neurons in the recurrent network back to the output).

Rather than constraining, as in *Equation (15)*, the low-dimensional input $\epsilon_\alpha$ and round-trip output $\sum_i d_{\alpha i} a_i(\vec{\epsilon})$ to be equal for each component $\alpha$ (expansive auto-encoder), we can alternatively enforce the high dimensional input $I_j$ (projection into neuron $j$ of low-dimensional input $\vec{\epsilon}$)

$$I_j \equiv \sum_\alpha e_{j\alpha} \epsilon_\alpha, \tag{16}$$

and round-trip output $I'_j \equiv \sum_{i,\alpha} e_{j\alpha} d_{\alpha i} \tilde{g}_i(I_i)$, where $\tilde{g}_i(I_i) \equiv a_i(\vec{\epsilon})$, to be equal for each neuron $j$ in the recurrent network (compressive auto-encoder) in order to optimize the decoding weights of the auto-encoder. Thus, the squared-error loss for this compressive auto-encoder becomes:

$$
\begin{aligned}
\mathcal{L}' &= \sum_p^P \sum_j^N \left( \sum_\alpha^{N_d} e_{j\alpha} \left( \sum_i^N d_{\alpha i} a_i^{(p)} - \epsilon_\alpha^{(p)} \right) \right)^2 \\
&= \sum_p^P \sum_j^N \left( \sum_\alpha^{N_d} e_{j\alpha} \left( \sum_i^N d_{\alpha i} a_i^{(p)} - \epsilon_\alpha^{(p)} \right) \right) \left( \sum_\gamma^{N_d} e_{j\gamma} \left( \sum_l^N d_{\gamma l} a_l^{(p)} - \epsilon_\gamma^{(p)} \right) \right) \\
&= \sum_p^P \sum_j^N \sum_\alpha^{N_d} e_{j\alpha}^2 \left( \sum_i^N d_{\alpha i} a_i^{(p)} - \epsilon_\alpha^{(p)} \right)^2 \\
&\qquad + \sum_p^P \sum_j^N \left( \sum_{\alpha,\gamma,\alpha \neq \gamma} e_{j\alpha} e_{j\gamma} \left( \sum_i^N d_{\alpha i} a_i^{(p)} - \epsilon_\alpha^{(p)} \right) \left( \sum_l^N d_{\gamma l} a_l^{(p)} - \epsilon_\gamma^{(p)} \right) \right) \\
&\approx c \sum_p^P \sum_\alpha^{N_d} \left( \sum_i^N d_{\alpha i} a_i^{(p)} - \epsilon_\alpha^{(p)} \right)^2,
\end{aligned}
\tag{17}
$$

where in the approximation, we exploit that (i) the relative importance of the term involving $\sum_p^P \sum_j \sum_{\alpha,\gamma,\alpha \neq \gamma} e_{j\alpha} e_{j\gamma}$ tends to zero as $1/\sqrt{NP}$, since $e_{j\alpha}$ and $e_{j\gamma}$ are independent random variables; and (ii) $\sum_j e_{j\alpha}^2 \approx c$ is independent of $\alpha$. Thus, the loss function of *Equation (17)* is approximately proportional to the squared-error loss function of *Equation (15)* (not considering the L2 regularization) used for the expansive auto-encoder, showing that for an auto-encoder embedded in a loop with fixed random encoding weights, the expansive and compressive descriptions are equivalent for those $N$-dimensional inputs $I_i$ that lie in the $N_d$-dimensional sub-space spanned by $\{e_{i\alpha}\}$ that is $I_i$ is of the form $\sum_\alpha e_{i\alpha} \epsilon_\alpha$ where $\epsilon_\alpha$ lies in a finite domain (hypersphere). We employed a large number $P = N$ of random low-$N_d$-dimensional inputs when constraining the expansive auto-encoder.

## Command input

The command input vector $\vec{u}(t)$ to the network was $N_c$-dimensional ($N_c = N_d$ for all systems except the arm) and time-varying. During the learning phase, input changed over two different time scales. The fast value of each command component was switched every 50 ms to a level $u'_\alpha$ chosen uniformly between $(-\zeta_1, \zeta_1)$ and this number was added to a more slowly changing input variable $\bar{u}_\alpha$ (called 'pedestal' in the main part of the paper) which changed with a period $T_{period}$. Here $\bar{u}_\alpha$ is the component of a vector of length $\zeta_2$ with a randomly chosen direction. The value of component $\alpha$ of the command is then $u_\alpha = \bar{u}_\alpha + u'_\alpha$. Parameter values for the network and input for each dynamical system are provided in *Table 1*. Further details are noted in the next subsection.

During the testing phase without error feedback, the network reproduced the reference trajectory of the dynamical system for a few seconds, in response to the same kind of input as during learning. We also tested the network on a different input not used during learning as shown in *Figures 2* and *4*.

## Equations and parameters for the example dynamical systems

The equations and input modifications for each dynamical system are detailed below. Time derivatives are in units of $s^{-1}$.

## Linear system

The equations for a linear decaying oscillator system (*Figure 2—figure supplement 2*) are

$$\dot{x}_1 = u_1/0.02 + (-0.2x_1 - x_2)/0.05$$
$$\dot{x}_2 = u_2/0.02 + (x_1 - 0.2x_2)/0.05.$$

For this linear dynamical system, we tested the learned network on a ramp of 2 s followed by a step to a constant non-zero value. A ramp can be viewed as a preparatory input before initiating an oscillatory movement, in a similar spirit to that observed in (pre-)motor cortex (*Churchland et al., 2012*). For such input too, the network tracked the reference for a few seconds (*Figure 2—figure supplement 2A–C*).

## van der Pol oscillator

The equations for the van der Pol oscillator system are

$$\dot{x}_1 = u_1/0.02 + x_2/0.125$$
$$\dot{x}_2 = u_2/0.02 + \left(2(1 - x_1^2)x_2 - x_1\right)/0.125.$$

Each component of the pedestal input $\bar{u}_\alpha$ was scaled differently for the van der Pol oscillator as reported in *Table 1*.

## Lorenz system

The equations for the chaotic Lorenz system (*Lorenz, 1963*) are

$$\dot{x}_1 = u_1/0.02 + 10(x_2 - x_1)$$
$$\dot{x}_2 = u_2/0.02 - x_1x_3 - x_2$$
$$\dot{x}_3 = u_3/0.02 + x_1x_2 - 8(x_3 + 28)/3.$$

In our equations above, $Z$ of the original Lorenz equations (*Lorenz, 1963*) is represented by an output variable $x_3 = Z - 28$ so as to have observable variables that vary around zero. This does not change the system dynamics, just its representation in the network. For the Lorenz system, only a pulse at the start for 250 ms, chosen from a random direction of norm $\zeta_1$, was provided to set off the system, after which the system followed autonomous dynamics.

## Non-linearly transformed input to linear system

For the above dynamical systems, the input adds linearly on the right hand sides of the differential equations. Our FOLLOW scheme also learned non-linear feedforward inputs to a linear dynamical system, as demonstrated in *Figure 2—figure supplement 4* and *Figure 2—figure supplement 5*. As the reference, we used the linear dynamical system above, but with its input transformed non-linearly by $g_\alpha(\vec{u}) = 10((u_\alpha/0.1)^3 - u_\alpha/0.4)$. Thus, the equations of the reference were:

$$\dot{x}_1 = 10((u_1/0.1)^3 - u_1/0.4) + (-0.2x_1 - x_2)/0.05$$
$$\dot{x}_2 = 10((u_2/0.1)^3 - u_2/0.4) + (x_1 - 0.2x_2)/0.05.$$

The input to the network remained $\vec{u}$. Thus, effectively the feedforward weights had to learn the non-linear transform $\vec{g}(\vec{u})$ while the recurrent weights learned the linear system.

## Arm dynamics

In the example of learning arm dynamics, we used a two-link model for an arm moving in the vertical plane with damping, under gravity (see for example http://www.gribblelab.org/compneuro/5_Computational_Motor_Control_Dynamics.html and https://github.com/studywolf/control/tree/master/studywolf_control/arms/two_link), with parameters from (*Li, 2006*). The differential equations for the four state variables, namely the shoulder and elbow angles $\vec{\theta} = (\theta_1, \theta_2)^T$ and the angular velocities $\vec{\omega} = (\omega_1, \omega_2)^T$, given input torques $\vec{\tau} = (\tau_1, \tau_2)^T$ are:

$$\dot{\vec{\theta}} = \vec{\omega} \tag{18}$$

$$\dot{\vec{\omega}} = M(\vec{\theta})^{-1}\left(\vec{\tau} - C(\vec{\theta},\vec{\omega}) - B\vec{\omega} - gD(\vec{\theta})\right), \tag{19}$$

with

$$M(\vec{\theta}) = \begin{pmatrix} d_1 + 2d_2\cos\theta_2 + m_1 s_1^2 + m_2 s_2^2 & d_3 + d_2\cos\theta_2 + m_2 s_2^2 \\ d_3 + d_2\cos\theta_2 + m_2 s_2^2 & d_3 + m_2 s_2^2 \end{pmatrix}$$

$$C(\vec{\theta},\vec{\omega}) = \begin{pmatrix} -\dot{\theta}_2(2\dot{\theta}_1 + \dot{\theta}_2) \\ \dot{\theta}_1^2 \end{pmatrix} d_2\sin\theta_2, B = \begin{pmatrix} b_{11} & b_{12} \\ b_{21} & b_{22} \end{pmatrix},$$

$$D(\vec{\theta}) = \begin{pmatrix} (m_1 s_1 + m_2 l_1)\sin\theta_1 + m_2 s_2\sin(\theta_1 + \theta_2) \\ m_2 s_2\sin(\theta_1 + \theta_2) \end{pmatrix},$$

$$d_1 = I_1 + I_2 + m_2 l_1^2, d_2 = m_2 l_1 s_2, d_3 = I_2,$$

where $m_i$ is the mass, $l_i$ the length, $s_i$ the distance from the joint centre to the centre of the mass, and $I_i$ the moment of inertia, of link $i$; $M$ is the moment of inertia matrix; $C$ contains centripetal and Coriolis terms; $B$ is for joint damping; and $D$ contains the gravitational terms. Here, the state variable vector $\vec{x} = [\theta_1, \theta_2, \omega_1, \omega_2]^T$, but the effective torque $\vec{\tau}$ is obtained from the input torque $\vec{u}$ as follows.

To avoid any link from rotating full 360 degrees, we provide an effective torque $\tau_\alpha$ to the arm, by subtracting a term proportional to the input torque $u_\alpha$, if the angle crosses $\pm 90$ degrees and $u_\alpha$ is in the same direction:

$$\tau_\alpha = u_\alpha - \begin{cases} u_\alpha\tilde{\sigma}(\theta_\alpha) & u_\alpha > 0 \\ 0 & u_\alpha = 0, \\ u_\alpha\tilde{\sigma}(-\theta_\alpha) & u_\alpha < 0 \end{cases}$$

where $\tilde{\sigma}(\theta)$ increases linearly from 0 to 1 as $\theta$ goes from $\pi/2$ to $3\pi/4$:

$$\tilde{\sigma}(\theta) = \begin{cases} 0 & \theta \leq \pi/2 \\ (\theta - \pi/2)/(\pi/4) & 3\pi/4 > \theta > \pi/2 \\ 1 & \theta \geq 3\pi/4 \end{cases}.$$

The parameter values were taken from the human arm (Model 1) in section 3.1.1 of the PhD thesis of Li (*Li, 2006*) from the Todorov lab; namely $m_1 = 1.4\,\text{kg}$, $m_2 = 1.1\,\text{kg}$, $l_1 = 0.3\,\text{m}$, $l_2 = 0.33\,\text{m}$, $s_1 = 0.11\,\text{m}$, $s_2 = 0.16\,\text{m}$, $I_1 = 0.025\,\text{kg}\,\text{m}^2$, $I_2 = 0.045\,\text{kg}\,\text{m}^2$, and $b_{11} = b_{22} = 0.05\,\text{kg}\,\text{m}^2/\text{s}$, $b_{12} = b_{21} = 0.025\,\text{kg}\,\text{m}^2/\text{s}$. Acceleration due to gravity was set at $g = 9.81\,\text{m/s}^2$. For the arm, we did not filter the reference variables for calculating the error.

The input torque $\vec{u}(t)$ for learning the two-link arm was generated, not by switching the pulse and pedestal values sharply, every 50 ms and $T_{period}$ as for the others, but by linearly interpolating in-between to avoid oscillations from sharp transitions.

The input torque $\vec{u}$ and the variables $\vec{\omega}$, $\vec{\theta}$ obtained on integrating the arm model above were scaled by 0.02 $(\text{Nm})^{-1}$, 0.05 $(\text{rad/s})^{-1}$ and $1/2.5$ $\text{rad}^{-1}$ respectively, and then these dimensionless variables were used as the input and reference for the spiking network. Effectively, we scaled the input torques to cover one-fifth of the representation radius $R_2$, the angular velocities one-half, and the angles full, as each successive variable was the integral of the previous one.

### Learning readout weights with recurrent weights fixed

For learning the readout weights after setting either the true or shuffled set of learned recurrent weights (*Figure 2—figure supplement 3*), we used a perceptron learning rule.

$$\frac{d}{dt}d_{\alpha i} = -\eta_r\left(\sum_j d_{\alpha j}(S_j * \kappa)(t) - x_\alpha(t)\right)(S_i * \kappa)(t) = -\eta_r(\hat{x}_\alpha(t) - x_\alpha(t))(S_i * \kappa)(t), \tag{20}$$

with learning rate $\eta_r = 1e-4$.

## Derivation and proof of stability of the FOLLOW learning scheme

We derive the FOLLOW learning rule *Equation (10)*, while simultaneously proving the stability of the scheme. We assume that: (1) the feedback $\{ke_{i\alpha}\}$ and readout weights $\{d_{\alpha j}\}$ form an auto-encoder

with gain $k$; (2) given the gains and biases of the spiking LIF neurons, there exist feedforward and recurrent weights that make the network follow the reference dynamics perfectly (in practice, the dynamics is only approximately realizable by our network, see Appendix 1 for a discussion); (3) the state $\vec{x}$ of the dynamical system is observable; (4) the intrinsic time scales of the reference dynamics are much larger than the synaptic time scale and the time scale of the error feedback loop, and much smaller than the time scale of learning; (5) the feedforward and recurrent weights remain bounded; and (6) the input $\vec{u}$ and reference output $\vec{x}$ remain bounded.

The proof proceeds in three major steps: (1) using the auto-encoder assumption to write the evolution equation of the low-dimensional output state variable in terms of the recurrent and feedforward weights; (2) showing that output follows the reference due to the error feedback loop; and (3) obtaining the evolution equation for the error and using it in the time-derivative of a Lyapunov function $V$, to show that $\dot{V} \leq 0$ for uniform stability, similar to proofs in adaptive control theory (**Narendra and Annaswamy, 1989**; **Ioannou and Sun, 2012**).

## Role of network weights for low-dimensional output

The filtered low-dimensional output of the recurrent network is given by **Equation (3)** of Results and repeated here:

$$\hat{x}_\alpha = \sum_j d_{\alpha j}(S_j * \kappa)(t),\tag{21}$$

where $d_{\alpha j}$ are the readout weights. Since $\kappa$ is an exponential filter with time constant $\tau_s$, **Equation (21)** can also be written as

$$\tau_s \dot{\hat{x}}_\alpha(t) = -\hat{x}_\alpha(t) + \sum_j d_{\alpha j} S_j(t),\tag{22}$$

We convolve this equation with kernel $\kappa$, multiply by the error feedback weights, and sum over the output components $\alpha$

$$\tau_s \sum_\alpha e_{i\alpha}(\dot{\hat{x}}_\alpha * \kappa)(t) = -\sum_\alpha e_{i\alpha}(\hat{x}_\alpha * \kappa)(t) + \sum_\alpha e_{i\alpha} \sum_j d_{\alpha j}(S_j * \kappa)(t).\tag{23}$$

We would like to write **Equation (23)** in terms of the recurrent and feedforward weights in the network.

To do this, we exploit assumptions (1) and (4). Having shown the equivalence of the compressive and expansive descriptions of our auto-encoder in the error-feedback loop (**Equations (15)** and **(17)**), we formulate our non-linear auto-encoder as compressive: we start with a high-dimensional set of inputs $I_j \equiv J_j - b_j$ (where $J_j$ is the current into neuron $j$ with bias $b_j$, cf. **Equations (5) and (6)**); transform these inputs non-linearly into filtered spike trains $S_j * \kappa$; decode these filtered spike trains into a low-dimensional representation $\vec{z}$ with components $z_\alpha = \sum_j d_{\alpha j}(S_j * \kappa)$; and increase the dimensionality back to the original one, via weights $ke_{i\alpha}$, to get inputs:

$$I_i' = \sum_\alpha ke_{i\alpha} z_\alpha = k \sum_\alpha \sum_j e_{i\alpha} d_{\alpha j}(S_j * \kappa).\tag{24}$$

Using assumption (1) we expect that the final inputs $I_i'$ are approximately $k$ times the initial inputs $I_i$:

$$k \sum_\alpha \sum_j e_{i\alpha} d_{\alpha j}(S_j * \kappa) \approx kI_i.\tag{25}$$

This is valid only if the high-$N$-dimensional input $I_i$ lies in the low-$N_d$-dimensional subspace spanned by $\{e_{i\alpha}\}$ (**Equation (17)**). We show that this requirement is fulfilled in the next major step of the proof (see text accompanying **Equations (31)–(35)**).

Our assumption (4) says that the state variables of the reference dynamics change slowly compared to neuronal dynamics. Due to the spatial averaging (sum over $j$ in **Equation (25)**) over a large number of neurons, individual neurons do not necessarily have to fire at a rate higher than the inverse of the synaptic time scale, while we can still assume that the total round trip input $I_i'$ on the

left hand side of *Equation (25)* is varying only on the slow time scale. Therefore, we used firing rate equations to compute mean outputs given static input when pre-calculating the readout weights (earlier in Methods).

Inserting the approximate *Equation (25)* into *Equation (23)* we find

$$\tau_s \sum_\alpha e_{i\alpha}(\dot{\hat{x}}_\alpha * \kappa)(t) \approx -\sum_\alpha e_{i\alpha}(\hat{x}_\alpha * \kappa)(t) + I_i(t). \tag{26}$$

We replace $I_i \equiv J_i - b_i$, using the current $J_i$ from *Equation (6)* for neuron $i$ of the recurrent network, to obtain

$$\tau_s \sum_\alpha e_{i\alpha}(\dot{\hat{x}}_\alpha * \kappa)(t) \approx -\sum_\alpha e_{i\alpha}(\hat{x}_\alpha * \kappa)(t) + \sum_j w_{ij}(S_j * \kappa)(t)$$
$$+ \sum_l w_{il}^{\mathrm{ff}}(S_l^{\mathrm{ff}} * \kappa)(t) + \sum_\alpha k e_{i\alpha}(\epsilon_\alpha * \kappa)(t). \tag{27}$$

Thus, the change of the low-dimensional output $\hat{x}_\alpha * \kappa$ depends on the network weights, which need to be learned. This finishes the first step of the proof.

## Error-feedback loop ensures that output follows reference

Because of assumption (2), we may assume that there exists a recurrent network of spiking neurons that represents the desired dynamics of *Equation (1)* without any error feedback. This second network serves as a target during learning and has variables and parameters indicated with an asterisk. In particular, the second network has feedforward weights $w_{il}^{\mathrm{ff}*}$ and recurrent weights $w_{ij}^*$. We write an equation similar to *Equation (27)* for the output $x_\alpha^*$ of the target network:

$$\tau_s \sum_\alpha e_{i\alpha}(\dot{x}_\alpha^* * \kappa)(t) = -\sum_\alpha e_{i\alpha}(x_\alpha^* * \kappa)(t) + \sum_j w_{ij}^*(S_j^* * \kappa)(t)$$
$$+ \sum_l w_{il}^{\mathrm{ff}*}(S_l^{\mathrm{ff}*} * \kappa)(t), \tag{28}$$

where $(S_l^{\mathrm{ff}*} * \kappa)(t)$ and $(S_j^* * \kappa)(t)$ are defined as the filtered spike trains of neurons in the realizable target network. We emphasize that this target network does not need error feedback because its output is, by definition, always correct. In fact, the readout from the spike trains $S_j^*$ gives the target output which we denote by $\vec{x}^*$. The weights of the target network are constant and their actual values are unimportant. They are mere mathematical devices to demonstrate stable learning of the first network which has adaptable weights. For the first network, we choose the same number of neurons and the same neuronal parameters as for the second network; moreover, the input encoding weights from the command input to the representation layer and the readout weights from the recurrent network to the output are identical for both networks. Thus, the only difference is that the feedforward and recurrent weights of the target network are realized, while for the first network they need to be learned.

In view of potential generalization, we note that any non-linear dynamical system is *approximately* realizable due to the expansion in a high-dimensional non-linear basis that is effectively performed by the recurrent network (see Appendix 1). Approximative weights (close to the ideal ones) could in principle also be calculated algorithmically (see Appendix 1). In the following we exploit assumption (2) and assume that the dynamics is actually (and not only approximately) realized by the target network.

Our assumption (3) states that the output is observable. Therefore the error component $\epsilon_\alpha$ can be computed directly via a comparison of the true output $\vec{x}$ of the reference with the output $\vec{\hat{x}}$ of the network: $\epsilon_\alpha = x_\alpha - \hat{x}_\alpha$. (In view of potential generalizations, we remark that the observable output need not be the state variables themselves, but could be a higher-dimensional non-linear function of the state variables, as shown for a general dynamical system in Appendix 1.)

As the second step of the proof, we now show that the error feedback loop enables the first network to follow the target network under assumptions (4 - 6). More precisely, we want to show that the readout and neural activities of the first network remain close to those of the target network at all times, that is $\hat{x}_\alpha(t) \approx x_\alpha^*(t)$ for each component $\alpha$ and $(S_i * \kappa)(t) \approx (S_i^* * \kappa)(t)$ for each neuron index $i$.

To do so, we use assumption (4) and exploit that (i) learning is slow compared to the network dynamics so the weights of the first network can be considered momentarily constant, and (ii) the reference dynamics is slower than the synaptic and feedback loop time scales, so the reference output $x_\alpha$ can be assumed momentarily constant. Thus, we have a separation of time scales in *Equation (27)*: for a given input (transmitted via the feedforward weights) and a given target value $x_\alpha^*$, the network dynamics settles on the fast time scale $\tau_s$ to a momentary fixed point $\hat{x}^\dagger$ which we find by setting the derivative on the left-hand side of *Equation (27)* to zero:

$$0 = -\sum_\alpha e_{i\alpha}(\hat{x}_\alpha^\dagger * \kappa)(t) + \sum_j w_{ij}(S_j * \kappa)(t) + \sum_l w_{il}^{\mathrm{ff}}(S_l^{\mathrm{ff}} * \kappa)(t) + \sum_\alpha k e_{i\alpha}((x_\alpha^* - \hat{x}_\alpha^\dagger) * \kappa)(t). \tag{29}$$

We rewrite this equation in the form

$$\sum_\alpha e_{i\alpha}(\hat{x}_\alpha^\dagger * \kappa)(t) = \frac{k}{k+1}\sum_\alpha e_{i\alpha}(x_\alpha^* * \kappa)(t) + \frac{1}{k+1}\left(\sum_j w_{ij}(S_j * \kappa)(t) + \sum_l w_{il}^{\mathrm{ff}}(S_l^{\mathrm{ff}} * \kappa)(t)\right). \tag{30}$$

We choose the feedback gain for the error much larger than 1 ($k \gg 1$), such that $k/(k+1) \approx 1$. We show below (in the text accompanying *Equations (31)–(35)*), that the factor in parentheses multiplying $1/(k+1)$ in the second term starts from zero and tends, with learning, towards $\sum_\alpha e_{i\alpha}(x_\alpha^* * \kappa)$, which is the factor multiplying $k/(k+1)$ in the first term. Thus, the first term remains approximately $k$ times larger than the second during learning. To obtain $\hat{x}_\alpha^\dagger \approx x_\alpha^*$, we set $k \gg 1$.

To show that the momentary fixed point is stable at the fast synaptic time scale, we calculate the Jacobian $\mathcal{J} = [\mathcal{J}_{il}]$, for the dynamical system given by *Equation (27)*. We introduce auxiliary variables $y_i \equiv \sum_\alpha e_{i\alpha}(\hat{x}_\alpha * \kappa)$ to rewrite *Equation (27)* with the new variables in the form $\dot{y}_i = F_i(\vec{y})$; and then we take derivative of its right hand side to obtain the elements of the Jacobian matrix at the fixed point $\sum_\alpha e_{i\alpha}\hat{x}_\alpha^\dagger$ (using $\int_0^\infty \kappa(\tau)d\tau = 1$):

$$\mathcal{J}_{il} \equiv \frac{\partial F_i(\vec{y})}{\partial y_l} = -\frac{(k+1)}{\tau_s}\delta_{il} + \frac{1}{\tau_s}\frac{\partial \sum_j w_{ij}(S_j * \kappa)(t)}{\partial y_l}\bigg|_{y_i = \sum_\alpha e_{i\alpha}\hat{x}_\alpha^\dagger}$$

where $\delta_{il}$ is the Kronecker delta function. We note that $\sum_j w_{ij}(S_j * \kappa)$ is a spatially and temporally averaged measure of the population activity in the network with appropriate weighting factors $w_{ij}$. We assume that the population activity varies smoothly with input, which is equivalent to requiring that on the time scale $\tau_s$, the network fires asynchronously, i.e. there are no precisely timed population spikes. Then we can take the second term to be bounded, in absolute value, by say $B_1/\tau_s$. The Jacobian matrix $\mathcal{J}$ is of the form $-(k+1)\mathbb{I}/\tau_s + \Lambda$, where $\mathbb{I}$ is the $N \times N$ identity matrix and $\Lambda$ is a matrix with each element bounded in absolute value by $B_1/\tau_s$. If we set $k \gg NB_1$, then all eigenvalues of the Jacobian have negative real parts, applying the Gerschgorin circle theorem (the second term can perturb any eigenvalue from $-(k+1)/\tau_s$ to within a circle of radius $NB_1/\tau_s$ at most), rendering the momentary fixed point asymptotically stable.

Thus, we have shown that if the initial state of the first network is close to the initial state of the target network, e.g., both start from rest, then on the slow time scale of the system dynamics of the reference $\vec{x}^*$, the first network output follows the target network output at all times, $\hat{x}_\alpha \approx x_\alpha^*$.

We now show that neurons are primarily driven by inputs close to those in the target network due to error feedback, and that these lie in the low-dimensional manifold spanned by $\{e_{i\alpha}\}$, as required for *Equation (25)*. We compute the projected error using *Equation (30)*:

$$\begin{aligned}\sum_\alpha e_{i\alpha}(\epsilon_\alpha * \kappa)(t) &= \sum_\alpha e_{i\alpha}((x_\alpha^* - \hat{x}_\alpha^\dagger) * \kappa)(t)\\ &= \frac{1}{k+1}\sum_\alpha e_{i\alpha}(x_\alpha^* * \kappa)(t) - \frac{1}{k+1}\left(\sum_j w_{ij}(S_j * \kappa)(t) + \sum_l w_{il}^{\mathrm{ff}}(S_l^{\mathrm{ff}} * \kappa)(t)\right),\end{aligned} \tag{31}$$

and insert it into *Equation (6)* to obtain the current into a neuron in the recurrent network:

$$J_i = \frac{k}{k+1}\sum_\alpha e_{i\alpha}(x_\alpha^* * \kappa)(t) + \frac{1}{k+1}\left(\sum_j w_{ij}(S_j * \kappa)(t) + \sum_l w_{il}^{\mathrm{ff}}(S_l^{\mathrm{ff}} * \kappa)(t)\right) + b_i \tag{32}$$

At the start of learning, if the feedforward and recurrent weights are small, then the neural input is dominated by the fed-back error input that is the first term, making $J_i$ close to the ideal current

$$J_i^* = \sum_\alpha e_{i\alpha}(x_\alpha^* * \kappa)(t) + b_i. \tag{33}$$

Thus, the neural input at the start of learning is of the form $\sum_\alpha e_{i\alpha} x_\alpha^*$ which lies in the low-dimensional subspace spanned by $\{e_{i\alpha}\}$ as required for **Equation (25)**. Furthermore, over time, the feedforward and recurrent weights get modified so that their contribution tends towards $\sum_\alpha e_{i\alpha}(x_\alpha^* * \kappa)$, such that the two terms of **Equation (32)** add to make $J_i$ even closer to the ideal current $J_i^*$ given by **Equation (33)**. This is made clearer by considering the weight update rule **Equation 10** as stochastic gradient descent on a loss function,

$$\mathcal{L}^J = \frac{1}{2} \sum_i \left( \sum_\alpha e_{i\alpha}(x_\alpha^* * \kappa)(t) - \sum_l w_{il}^{\text{ff}}(S_l^{\text{ff}} * \kappa)(t) - \sum_j w_{ij}(S_j * \kappa)(t) \right)^2, \tag{34}$$

leading us to (for each recurrent weight $w_{ij}$, and similarly for $w_{il}^{\text{ff}}$):

$$\begin{aligned}
\dot{w}_{ij} &= -\eta' \frac{\partial \mathcal{L}^J}{\partial w_{ij}} \\
&= \eta' \left( \sum_\alpha e_{i\alpha}(x_\alpha^* * \kappa) - \sum_l w_{il}^{\text{ff}}(S_l^{\text{ff}} * \kappa) - \sum_j w_{ij}(S_j * \kappa) \right)(S_j * \kappa) \\
&= \eta' \frac{k+1}{k}(I_i^\epsilon * \kappa)(S_j * \kappa),
\end{aligned} \tag{35}$$

which is identical to the FOLLOW learning rule for $w_{ij}$ in **Equation (10)** except for the time-scale of filtering of the error current (see Discussion), and a factor involving $k$ that can be absorbed into the learning rate $\eta'$. In the last step above, we used the projected error current from **Equation (31)** and the definition of $I_i^\epsilon$ in **Equation (8)**. Thus, the feedforward and recurrent connections evolve to inject, after learning, the same ideal input within the low-dimensional manifold, as was provided by the error feedback during learning. Hence, the neural input remains in the low-dimensional manifold spanned by $\{e_{i\alpha}\}$ throughout learning, as required for **Equation (25)**, making this major step and the previous one self-consistent.

Since the driving neural currents are close to ideal throughout learning, the filtered spike trains of the recurrent neurons in the first network will also be approximately the same as those of the target network, so that $(S_i * \kappa)(t)$ can be used instead of $(S_i^* * \kappa)(t)$ in (**Equation (28)**). Moreover, the filtered spike trains $(S_l^{\text{ff}} * \kappa)(t)$ of the command representation layer in the first network are the same as those in the target network, since they are driven by the same command input $\vec{u}$ and the command encoding weights are, by construction, the same for both networks. The similarity of the spike trains in the first and target networks will be used in the next major part of the proof.

## Stability of learning via Lyapunov's method

We now turn to the third step of the proof and consider the temporal evolution of the error $\epsilon_\alpha = x_\alpha - \hat{x}_\alpha$. We exploit that the network dynamics is realized by the target network and insert **Equations (27) and (28)** so as to find

$$\begin{aligned}
-\tau_s \sum_\alpha e_{i\alpha}(\dot{\epsilon}_\alpha * \kappa)(t) &= \tau_s \sum_\alpha e_{i\alpha}((\dot{\hat{x}}_\alpha - \dot{x}_\alpha) * \kappa)(t) \\
&\approx \tau_s \sum_\alpha e_{i\alpha}((\dot{\hat{x}}_\alpha - \dot{x}_\alpha^*) * \kappa)(t) \\
&\approx \sum_j \left( w_{ij} - w_{ij}^* \right)(S_j * \kappa)(t) + \sum_l \left( w_{il}^{\text{ff}} - w_{il}^{\text{ff}*} \right)(S_l^{\text{ff}} * \kappa)(t) \\
&\quad + (k+1) \sum_\alpha e_{i\alpha}(\epsilon_\alpha * \kappa)(t) \\
&\equiv \sum_j \psi_{ij}(S_j * \kappa)(t) + \sum_l \phi_{il}(S_l^{\text{ff}} * \kappa)(t) + (k+1) \sum_\alpha e_{i\alpha}(\epsilon_\alpha * \kappa)(t).
\end{aligned} \tag{36}$$

In the second line, we have replaced the reference output by the target network output; and in the third line we have used **Equations (27) and (28)**, and replaced the filtered spike trains of the target network by those of the first network, exploiting the insights from the previous paragraph. In the last line, we have introduced abbreviations $\psi_{ij} \equiv w_{ij} - w_{ij}^*$ and $\phi_{il} \equiv w_{il}^{\text{ff}} - w_{il}^{\text{ff}*}$.

In order to show that the absolute value of the error decreases over time with an appropriate learning rule, we consider the candidate Lyapunov function:

$$V(\tilde{\epsilon}, \psi, \phi) = \frac{1}{2} \sum_i \tilde{\epsilon}_i^2 + \frac{1}{2} \frac{1}{\tilde{\eta}_1} \sum_{i,j} (\psi_{ij})^2 + \frac{1}{2} \frac{1}{\tilde{\eta}_2} \sum_{i,l} (\phi_{il})^2,$$

(37)

where $\tilde{\epsilon}_i \equiv \tau_s \sum_\alpha e_{i\alpha} (\epsilon_\alpha * \kappa)$ and $\tilde{\eta}_1, \tilde{\eta}_2 > 0$ are positive constants. We use Lyapunov's direct method to show the stability of learning. For this, we require the following properties for the Lyapunov function. (a) The Lyapunov function is positive semi-definite $V(\tilde{\epsilon}, \psi, \phi) \geq 0$, with the equality to zero only at $(\tilde{\epsilon}, \psi, \phi) = (0, 0, 0)$. (b) It has continuous first-order partial derivatives. Furthermore, $V$ is (c) *radially unbounded* since

$$V(\tilde{\epsilon}, \psi, \phi) > |(\tilde{\epsilon}, \psi, \phi)|^2 / (4 \max(1, \tilde{\eta}_1, \tilde{\eta}_2)),$$

and (d) *decrescent* since

$$V(\tilde{\epsilon}, \psi, \phi) < |(\tilde{\epsilon}, \psi, \phi)|^2 / \min(1, \tilde{\eta}_1, \tilde{\eta}_2),$$

where $|(\tilde{\epsilon}, \psi, \phi)|^2 \equiv \sum_i (\tilde{\epsilon}_i)^2 + \sum_{i,j} (\psi_{ij})^2 + \sum_{i,k} (\phi_{il})^2$ and $\min / \max$ take the minimum/maximum of their respective arguments.

Apart from the above conditions (a)-(d), we need to show the key property $\dot{V} \leq 0$ for uniform global stability (which implies that bounded orbits remain bounded, so the error remains bounded); or the stronger property $\dot{V} < 0$ for asymptotic global stability (see for example [**Narendra and Annaswamy, 1989**; **Ioannou and Sun, 2012**]). Taking the time derivative of $V$, and replacing $\dot{\tilde{\epsilon}}_i$ that is $\tau_s \sum_\alpha e_{i\alpha} (\dot{\epsilon}_\alpha * \kappa)$ from (**Equation (36)**), we have:

$$
\begin{aligned}
\dot{V} &= \sum_i \tilde{\epsilon}_i \dot{\tilde{\epsilon}}_i + \frac{1}{\tilde{\eta}_1} \sum_{i,j} \psi_{ij} \dot{\psi}_{ij} + \frac{1}{\tilde{\eta}_2} \sum_{i,l} \phi_{il} \dot{\phi}_{il} \\
&\approx -\sum_i \tilde{\epsilon}_i \left( \sum_j \psi_{ij} (S_j * \kappa)(t) + \sum_l \phi_{il} (S_l^{\text{ff}} * \kappa)(t) + (k+1) \sum_\alpha e_{i\alpha} (\epsilon_\alpha * \kappa)(t) \right) \\
&\quad + \frac{1}{\tilde{\eta}_1} \sum_{i,j} \psi_{ij} \dot{\psi}_{ij} + \frac{1}{\tilde{\eta}_2} \sum_{i,l} \phi_{il} \dot{\phi}_{il} \\
&= \sum_{i,j} \psi_{ij} \left( -\tilde{\epsilon}_i (S_j * \kappa)(t) + \frac{1}{\tilde{\eta}_1} \dot{\psi}_{ij} \right) \\
&\quad + \sum_{i,k} \phi_{il} \left( -\tilde{\epsilon}_i (S_l^{\text{ff}} * \kappa)(t) + \frac{1}{\tilde{\eta}_2} \dot{\phi}_{il} \right) - (k+1) \sum_i \tilde{\epsilon}_i^2 / \tau_s.
\end{aligned}
$$

(38)

If we enforce the first two terms above to be zero, we derive a learning rule

$$
\begin{aligned}
\dot{\psi}_{ij} &= \tilde{\eta}_1 \tilde{\epsilon}_i (S_j * \kappa)(t) \\
\dot{\phi}_{il} &= \tilde{\eta}_2 \tilde{\epsilon}_i (S_l^{\text{ff}} * \kappa)(t),
\end{aligned}
$$

(39)

and then

$$\dot{V} = -(k+1) \sum_i \tilde{\epsilon}_i^2 / \tau_s \leq 0$$

requiring $k > -1$, which is subsumed under $k \gg 1$ for the error feedback. **Equation (39)** with $\eta_1 \equiv \tilde{\eta}_1 \tau_s / k$ and $\eta_2 \equiv \tilde{\eta}_2 \tau_s / k$, and $\kappa$ replaced by a longer filtering kernel $\kappa^\epsilon$, is the learning rule used in the main text, **Equation (10)**.

Thus, in the $(\tilde{\epsilon}, \psi, \phi)$-system given by **Equations (36) and (39)**, we have proven the global uniform stability of the fixed point $(\tilde{\epsilon}, \psi, \phi) = (0, 0, 0)$, which is effectively $(\epsilon, \psi, \phi) = (0, 0, 0)$, choosing $\eta_1, \eta_2 > 0$

and $k \gg \max(1, NB_1)$, under assumptions (1 - 6), while simultaneously deriving the learning rule (*Equation (39)*).

This ends our proof. So far, we have shown that the system is Lyapunov stable, that is bounded orbits remain bounded, and not asymptotically stable. Indeed, with bounded firing rates and fixed readout weights, the output will remain bounded, as will the error (for a bounded reference). However, here, we also derived the FOLLOW learning rule, and armed with the inequality for the time derivative of the Lyapunov function in terms of the error, we further show in the following subsection that the error $\vec{\epsilon}$ goes to zero asymptotically, so that after learning, even without error feedback, $\hat{\vec{x}}$ reproduces the dynamics of $\vec{x}$.

A major caveat of this proof is that under assumption (2) the dynamics be *realizable* by our network. In a real application this might not be the case. Approximation errors arising from a mismatch between the best possible network and the actual target dynamics are currently ignored. The adaptive control literature has shown that errors in approximating the reference dynamics appear as frozen noise and can cause runaway drift of the parameters (*Narendra and Annaswamy, 1989*; *Ioannou and Sun, 2012*). In our simulations with a large number of neurons, the approximations of a non-realizable reference dynamics (e.g., the van der Pol oscillator) were sufficiently good, and thus the expected drift was possibly slow, and did not cause the error to rise during typical time-scales of learning. A second caveat is our assumption (5). While the input is under our control and can therefore be kept bounded, some additional bounding is needed to stop weights from drifting. Various techniques to address such model-approximation noise and bounding weights have been studied in the robust adaptive control literature (e.g., (*Ioannou and Tsakalis, 1986*; *Slotine and Coetsee, 1986*; *Narendra and Annaswamy, 1989*; *Ioannou and Fidan, 2006*; *Ioannou and Sun, 2012*)). We discuss this issue and briefly mention some of these ameliorative techniques in Appendix 1.

To summarize, the FOLLOW learning rule (*Equation (39)*) on the feedforward or recurrent weights has two terms: (i) a filtered presynaptic firing trace $(S_i^{\text{ff}} * \kappa)(t)$ or $(S_j * \kappa)(t)$ that is available locally at each synapse; and (ii) a projected filtered error $\sum_\alpha k e_{i\alpha}(\epsilon_\alpha * \kappa)(t)$ used for all synapses in neuron $i$ that is available as a current in the postsynaptic neuron $i$ due to error feedback, see *Equation (6)*. Thus the learning rule can be classified as local. Moreover, it uses an error in the observable $\vec{x}$, not in its time-derivative. While we have focused on spiking networks, the learning scheme can be easily used for non-linear rate units by replacing the filtered spikes $(S_i * \kappa)(t)$ by the output of the rate units $r(t)$. Our proof is valid for arbitrary dynamical transforms $\vec{h}(\vec{x}, \vec{u})$ as long as they are realizable in a network. The proof shows uniform global stability using Lyapunov's method.

## Proof of error tending to zero asymptotically

In the above subsection, we showed uniform global stability using $\dot{V} = -(k+1)\sum_i (\tilde{\epsilon}_i)^2 \leq 0$, with $k \gg \max(1, NB_1)$ and $\tilde{\epsilon}_i \equiv \tau_s \sum_\alpha e_{j\alpha}(\epsilon_\alpha * \kappa)$. This only means that bounded errors remain bounded. Here, we show more importantly that the error tends to zero asymptotically with time. We adapt the proof in section 4.2 of (*Ioannou and Sun, 2012*), to our spiking network.

Here, we want to invoke a special case of Barbălat's lemma: if $f, \dot{f} \in \mathcal{L}_\infty$ and $f \in \mathcal{L}_p$ for some $p \in [1, \infty)$, then $f(t) \to 0$ as $t \to \infty$. Recall the definitions: function $f \in \mathcal{L}_p$ when $||x||_p \equiv \left(\int_0^\infty |f(\tau)|^p d\tau\right)^{1/p}$ exists (is finite); and similarly function $f \in \mathcal{L}_\infty$ when $||x||_\infty \equiv \sup_{t \geq 0} |f(t)|$ exists (is finite).

Since $V$ is positive semi-definite ($V \geq 0$) and is a non-increasing function of time ($\dot{V} \leq 0$), its $\lim_{t\to\infty} V = V_\infty$ exists and is finite. Using this, the following limit exists and is finite:

$$\sum_i \int_0^\infty (\tilde{\epsilon}_i(\tau))^2 d\tau = \frac{-1}{k+1}\int_0^\infty \dot{V}(\tau)d\tau = \frac{1}{k+1}(V(0) - V_\infty).$$

Since each term in the above sum $\sum_i$ is positive semi-definite, $\int_0^\infty (\tilde{\epsilon}_i(\tau))^2 d\tau$ also exists and is finite $\forall i$, and thus $\tilde{\epsilon}_i \in \mathcal{L}_2 \forall i$.

To show that $\tilde{\epsilon}_i, \dot{\tilde{\epsilon}}_i \in \mathcal{L}_\infty \forall i$, consider *Equation (36)*. We use assumption (6) that the input $\vec{u}(t)$ and the reference output $\vec{x}(t)$ are bounded. Since network output $\hat{\vec{x}}$ is also bounded due to saturation of firing rates (as are the filtered spike trains), the error (each component) is bounded that is $\tilde{\epsilon}_i \in \mathcal{L}_\infty \forall i$. If we also bound the weights from diverging during learning (assumption (5)), then $\psi_{ij}, \phi_{il} \in \mathcal{L}_\infty \forall i, j, l$.

With these reasonable assumptions, all terms on the right hand side of the *Equation (36)* for $\dot{\tilde{\epsilon}}_i$ are bounded, hence $\dot{\tilde{\epsilon}}_i \in \mathcal{L}_\infty \, \forall i$.

Since $\tilde{\epsilon}_i \in \mathcal{L}_2 \, \forall i$ and $\tilde{\epsilon}_i, \dot{\tilde{\epsilon}}_i \in \mathcal{L}_\infty \, \forall i$, invoking Barbălat's lemma as above, we have $\tilde{\epsilon}_i \to 0 \, \forall i$ as $t \to \infty$. We have shown that the error tends to zero asymptotically under assumptions (1 - 6). In practice, the error shows fluctuations on a short time scale while the mean error over a longer time scale reduces and then plateaus, possibly due to approximate realizability, imperfections in the error-feedback, and spiking shot noise (cf. *Figure 5*).

We do not further require the convergence of parameters to ideal ones for our purpose, since the error tending to zero, that is network output matching reference, is functionally sufficient for the forward predictive model. In the adaptive control literature (*Ioannou and Sun, 2012*; *Narendra and Annaswamy, 1989*), the parameters (weights) are shown to converge to ideal ones if input excitation is 'persistent', loosely that it excites all modes of the system. It should be possible to adapt the proof to our spiking network, as suggested by simulations (*Figure 5*), but is not pursued here.

## Acknowledgements

We thank Johanni Brea, Samuel Muscinelli and Laureline Logiaco for helpful discussions, and Samuel Muscinelli, Laureline Logiaco, Chris Stock, Tilo Schwalger, Olivia Gozel, Dane Corneil and Vasiliki Liakoni for comments on the manuscript. Financial support was provided by the European Research Council (Multirules, grant agreement no. 268689), the Swiss National Science Foundation (Sinergia, grant agreement no. CRSII2_147636), and the European Commission Horizon 2020 Framework Program (H2020) (Human Brain Project, grant agreement no. 720270).

## Additional information

### Funding

| Funder | Grant reference number | Author |
| --- | --- | --- |
| European Research Council | Multirules 268 689 | Aditya Gilra<br>Wulfram Gerstner |
| Schweizerischer Nationalfonds zur Förderung der Wissenschaftlichen Forschung | CRSII2_147636 | Aditya Gilra<br>Wulfram Gerstner |
| Horizon 2020 Framework Programme | Human Brain Project 720270 | Wulfram Gerstner |

The funders had no role in study design, data collection and interpretation, or the decision to submit the work for publication.

### Author contributions

Aditya Gilra, Conceptualization, Data curation, Software, Formal analysis, Investigation, Visualization, Methodology, Writing—original draft, Writing—review and editing; Wulfram Gerstner, Conceptualization, Resources, Supervision, Funding acquisition, Writing—review and editing

### Author ORCIDs

Aditya Gilra http://orcid.org/0000-0002-8628-1864

### Decision letter and Author response

Decision letter https://doi.org/10.7554/eLife.28295.022
Author response https://doi.org/10.7554/eLife.28295.023

## Additional files

### Supplementary files

• Transparent reporting form

DOI: https://doi.org/10.7554/eLife.28295.019

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

## Appendix 1

DOI: https://doi.org/10.7554/eLife.28295.020

### Decoding

Consider only the command representation layer without the subsequent recurrent network. Assume, following (*Eliasmith and Anderson, 2004*), we wish to decode an arbitrary output $\vec{v}(\vec{u})$ corresponding to the $\vec{u}$ encoded in the command representation layer, from the spike trains $S_l^{\text{ff}}(t)$ of the neurons, by synaptically filtering and linearly weighting the trains with decoding weights $d_{\alpha l}^{(\vec{v})}$:

$$\hat{v}_\alpha(\vec{u}) = \sum_l d_{\alpha l}^{(\vec{v})} (S_l^{\text{ff}} * \kappa)(t), \tag{40}$$

where $*$ denotes convolution $(S_l^{\text{ff}} * \kappa)(t) \equiv \int_{-\infty}^{t} S_l^{\text{ff}}(t')\kappa(t-t')dt' = \int_0^\infty S_l^{\text{ff}}(t-t')\kappa(t')dt'$, and $\kappa(t) \equiv \exp(-t/\tau_s)/\tau_s$ is a normalized filtering kernel.

We can obtain the decoders $d_{\alpha i}^{(\vec{v})}$ by minimizing the loss function

$$L = \left\langle \sum_\alpha \left( v_\alpha(\vec{u}) - \sum_l d_{\alpha l}^{(\vec{v})} \langle S_l^{\text{ff}} * \kappa \rangle_t \right)^2 \right\rangle_{\vec{u}} \tag{41}$$

with respect to the decoders. The average $\langle \cdot \rangle_{\vec{u}}$ over $\vec{u}$ guarantees that the same constant decoders are used over the whole range of constant inputs $\vec{u}$. The time average $\langle \cdot \rangle_t$ denotes an analytic rate computed for each constant input for a LIF neuron. Linear regression with a finite set of constant inputs $\vec{u}$ was used to obtain the decoders (see Methods). With these decoders, if the input $\vec{u}$ varies slowly compared to the synaptic time constant $\tau_s$, we have $\hat{v}_\alpha = \sum_l d_{\alpha l}^{(\vec{v})}(S_l^{\text{ff}} * \kappa)(t) \approx v_\alpha(\vec{u})$.

Any function of the input $\vec{v}(\vec{u})$ can be approximated with appropriate linear decoding weights $d_{\alpha l}^{(\vec{v})}$ from the high-dimensional basis of non-linear tuning curves of heterogeneous neurons with different biases, encoding weights and gains, schematized in *Figure 1—figure supplement 1*. With a large enough number of such neurons, the function is expected to be approximated to arbitrary accuracy. While this has not been proven rigorously for spiking neurons, this has theoretical underpinnings from theorems on universal function approximation using non-linear basis functions (*Funahashi, 1989*; *Hornik et al., 1989*; *Girosi and Poggio, 1990*) successful usage in spiking neural network models by various groups (*Seung et al., 2000*; *Eliasmith and Anderson, 2004*; *Eliasmith, 2005*), and biological plausibility (*Poggio, 1990*; *Burnod et al., 1992*; *Pouget and Sejnowski, 1997*).

Here, the neurons that are active at any given time operate in the mean driven regime, that is the instantaneous firing rate increases with the input current (*Gerstner et al., 2014*). The dynamics is dominated by synaptic filtering, and the membrane time constant does not play a significant role (*Eliasmith and Anderson, 2004*; *Eliasmith, 2005*; *Seung et al., 2000*; *Abbott et al., 2016*). Thus, the decoding weights derived from *Equation (41)* with stationary input are good approximations even in the time-dependent case, as long as the input varies on a time scale slower than the synaptic time constant.

### Online learning based on a loss function and its shortcomings

Suppose that a dynamical system given by

$$\dot{x}_\alpha = f_\alpha(\vec{x}) + g_\alpha(\vec{u}) \tag{42}$$

is to be mimicked by our spiking network implementing a different dynamical system with an extra error feedback term as in *Equation (27)*. This can be interpreted as:

$$\tau_s \dot{\hat{x}}_\alpha = -\hat{x}_\alpha + \breve{f}_\alpha(\vec{\hat{x}}, \vec{u}) + \breve{g}_\alpha(\vec{u}) + k\epsilon_\alpha. \tag{43}$$

Comparing with the reference *Equation (42)*, after learning we want that $\breve{f}_\alpha(\vec{\hat{x}}, \vec{u}) + \breve{g}_\alpha(\vec{u})$ should approximate $\tau_s f_\alpha(\vec{\hat{x}}) + \hat{x}_\alpha + \tau_s g_\alpha(\vec{u})$. One way to achieve this (*Eliasmith and Anderson, 2004*) is to ensure that $\breve{f}_\alpha(\vec{\hat{x}}, \vec{u})$ and $\breve{g}_\alpha(\vec{u})$ approximate $\tilde{f}_\alpha(\vec{\hat{x}}) \equiv \tau_s f_\alpha(\vec{\hat{x}}) + \hat{x}_\alpha$ and $\tilde{g}_\alpha(\vec{u}) \equiv \tau_s g_\alpha(\vec{u})$ respectively, as used in the loss functions below. In our simulations, we usually start with zero feedforward and recurrent weights, so that initially $\breve{f}(\vec{\hat{x}}, \vec{u}) = 0 = \breve{g}_\alpha(\vec{u})$.

Assuming that the time scales of dynamics are slower than synaptic time scale $\tau_s$, we can approximate the requisite feedforward and recurrent weights, by minimizing the following loss functions respectively, with respect to the weights (*Eliasmith and Anderson, 2004*):

$$L_{\text{ff}} = \left\langle \sum_j \left( \sum_\alpha e_{k\alpha}^{\text{ff}} \tilde{g}_\alpha(\vec{u}) - \sum_l w_{jl} \langle S_l^{\text{ff}} * \kappa \rangle_t \right)^2 \right\rangle_x, \tag{44}$$

$$L_{rec} = \left\langle \sum_j \left( \sum_\alpha e_{j\alpha} \tilde{f}_\alpha(\vec{x}) - \sum_i w_{ji} \langle S_i * \kappa \rangle_t \right)^2 \right\rangle_x. \tag{45}$$

Using these loss functions, we can pre-calculate the weights required for any dynamical system numerically, similarly to the calculation of decoders in the subsection above.

We now derive rules for learning the weights online based on stochastic gradient descent of these loss functions, similar to (*MacNeil and Eliasmith, 2011*), and point out some shortcomings.

The learning rule for the recurrent weights by gradient descent on the loss function given by *Equation (45)* is

$$
\begin{aligned}
\frac{dw_{ji}}{dt} &= -\frac{1}{2}\eta \frac{\partial L_{rec}}{\partial w_{ji}} \\
&\approx \eta \left\langle \left( \sum_\alpha e_{j\alpha} \tilde{f}_\alpha(\vec{x}) - \sum_i w_{ji}(S_i * \kappa)(t) \right)(S_i * \kappa)(t) \right\rangle_x \\
&\equiv \eta \left\langle \epsilon_j^{(\tilde{f})}(S_i * \kappa)(t) \right\rangle_x.
\end{aligned}
\tag{46}
$$

In the second line, the effect of the weight change on the filtered spike trains is assumed small and neglected, using a small learning rate $\eta$. With requisite dynamics slower than synaptic $\tau_s$, and with large enough number of neurons, we have approximated $\sum_i w_{ji} \langle S_i * \kappa \rangle_t(t) \approx \sum_i w_{ji}(S_i * \kappa)(t)$. The third line defines an error in the projected $\vec{\tilde{f}}(\vec{x})$, which is the supervisory signal.

If we assume that the learning rate is slow, and the input samples the range of $x$ uniformly, then we can remove the averaging over $x$, similar to stochastic gradient descent

$$\frac{dw_{ji}}{dt} \approx \eta \epsilon_j^{(\tilde{f})}(S_i * \kappa)(t), \tag{47}$$

where $\epsilon_j^{(\tilde{f})} \equiv \left( \sum_\alpha e_{j\alpha} \tilde{f}_\alpha(\vec{x}) - \sum_i w_{ji}(S_i * \kappa)(t) \right)$. This learning rule is the product of a multi-dimensional error $\epsilon_j^{(\tilde{f})}$ and the filtered presynaptic spike train $(S_i * \kappa)(t)$. However, this error in the unobservable $\vec{\tilde{f}}$ is not available to the postsynaptic neuron, making the learning rule non-local. A similar issue arises in the feedforward case.

In mimicking a dynamical system, we want only the observable output of the dynamical system, that is $\vec{x}$ to be used in a supervisory signal, not a term involving the unknown $\vec{f}(\vec{x})$ that appears in the derivative $\dot{\vec{x}}$. Even if this derivative is computed from the observable $\vec{x}$, it will be noisy. Furthermore, this derivative cannot be obtained by differentiating the observable versus time, if the observable is not directly the state variable, but an unknown non-linear function of it, which however our FOLLOW learning can handle

(see next subsection). Thus, this online rule, if using just the observable error, can learn only an integrator for which $f(x) \sim x$ (*MacNeil and Eliasmith, 2011*).

Indeed, learning both the feedforward and recurrent weights simultaneously using gradient descent on these loss functions, requires two different and unavailable error currents to be projected into the postsynaptic neuron to make the rule local.

## General dynamical system and transformed observable

General dynamical systems of the form

$$\frac{d\vec{x}(t)}{dt} = \vec{h}(\vec{x}(t), \vec{u}(t)),$$
$$\vec{y}(t) = \vec{K}(\vec{x}(t))$$

can be learned with the same network configuration (*Figure 1B*) used for systems of the form *Equation 1*. Here, the state variable is $\vec{x}$, but the observable which serves as the reference to the network is $\vec{y}$. The transformation equation of the observable (second equation) can be absorbed into the first equation as below.

Consider the transformation equation for the observable. The dimensionality of the relevant variables: (1) the state variables (say joint angles and velocities) $\vec{x}$; (2) the observables represented in the brain (say sensory representations of the joint angles and velocities) $\vec{y}$; and (3) the control input (motor command) $\vec{u}$, can be different from each other, but must be small compared to the number of neurons. Furthermore, we require the observable $\vec{y}$ to not lose information compared to $\vec{x}$, that is $\vec{K}$ must be invertible, so $\vec{y}$ will have at least the same dimension as $\vec{x}$.

The time evolution of the observable is

$$\dot{y}_\beta = \sum_\alpha \frac{\partial K_\beta(\vec{x})}{\partial x_\alpha} \dot{x}_\alpha = \sum_\alpha \frac{\partial K_\beta(\vec{x})}{\partial x_\alpha} h_\alpha(\vec{x}, \vec{u}) \equiv p_\beta(\vec{y}, \vec{u}).$$

The last step follows since function $\vec{K}$ is invertible, so that $\vec{x} = \vec{K}^{-1}(\vec{y})$. So we essentially need to learn $\dot{y}_\beta = p_\beta(\vec{y}, \vec{u})$.

Having solved the observable transformation issue, we use $\vec{x}$ now for our observable instead of $\vec{y}$, consistent with the main text. The dynamical system to be learned is now $\dot{x}_\beta = p_\beta(\vec{x}, \vec{u})$. Since our learning network effectively evolves as *Equation (43)*, it can approximate $p_\beta(\vec{x}, \vec{u})$. Thus our network can learn general dynamical systems with observable transformations.

## Approximation error causes drift in weights

A frozen noise term $\xi(\vec{x}(t))$ due to the approximate decoding from non-linear tuning curves of neurons, by the feedforward weights, recurrent weights and output decoders, will appear additionally in *Equation (36)*. If this frozen noise has a non-zero mean over time as $\vec{x}(t)$ varies, leading to a non-zero mean error, then it causes a drift in the weights due to the error-based learning rules in *Equation (10)*, and possibly a consequent increase in error. Note that the stability and error tending to zero proofs assume that this frozen noise is negligible.

Multiple strategies with contrasting pros and cons have been proposed to counteract this parameter drift in the robust adaptive control literature (*Ioannou and Sun, 2012*; *Narendra and Annaswamy, 1989*; *Ioannou and Fidan, 2006*). These include a weight leakage/regularizer term switched slowly on, when a weight crosses a threshold (*Ioannou and Tsakalis, 1986*; *Narendra and Annaswamy, 1989*), or a dead zone strategy with no updating of weights once the error is lower than a set value (*Slotine and Coetsee, 1986*; *Ioannou and Sun, 2012*). In our simulations, the error continued to drop even over longer than typical learning time scales (*Figure 5*), and so, we did not implement these strategies.

In practice, the learning can be stopped once error is low enough, while the error feedback can be continued, so that the learned system does not deviate too much from the observed one.

