## [Decision Letter]

Thank you for submitting your article "Predicting non-linear dynamics by stable local learning in a recurrent spiking neural network" for consideration by *eLife*. Your article has been reviewed by three peer reviewers, one of whom is a member of our Board of Reviewing Editors and the evaluation has been overseen by Timothy Behrens as the Senior Editor.

The reviewers have discussed the reviews with one another and the Reviewing Editor has drafted this decision to help you prepare a revised submission.

Summary:

This paper proposes a learning scheme that allows a recurrent network of spiking neurons to predict the dynamics of a non-linear system by training feedforward and recurrent connections. The learning is achieved using a local learning rule. The error signal is fed back by fixed, random connections, with the feedback loop forming an auto-encoder. After learning, which is done via motor "babbling," the network is able to reproduce the output in an open-loop setting without feedback. The authors showcase their training algorithm in a number of linear and non-linear applications, including the dynamics of a two-link arm. In addition, the authors prove analytically under mild assumptions that learning converges if the non-linear dynamics can be realized by the network. The paper is mainly clearly written, and has the potential to be of significant interest to the community.

Essential revisions:

1) The firing rates are far higher than is biologically plausible. It is important that this is fixable, at least in principle. If not, then the scheme has nothing to do with the brain. We hesitate to ask for additional simulations, but we think you need to show that the network can work when firing rates are in a more biologically plausible range, as in less than about 20 Hz on average (still high, but closer to reality).

2) We did not follow a key step. The sixth equation in subsection “Network architecture and parameters” says:

sum_{beta, i} e_{j,beta} d_{beta,i} a_i(sum_alpha e_{i,alpha} epsilon_alpha)

\approx sum_alpha e_{j,alpha} epsilon_alpha

It seems that you effectively rewrote this as

sum_{beta, i} e_{j,beta} d_{beta,i} a_i(I_i) \approx I_i

(see Eq. (13)). Strictly speaking, this holds only if the current (whose ith component is I_i) lies in the low dimensional submanifold spanned by the e_{i,alpha}. However, because of the recurrent dynamic, the network is likely be chaotic, and so the current is likely to live on a high-dimensional manifold.

This seems to indicate a problem with the derivation. Or are we missing something? In any case, this needs to be explained much more clearly.

3) It would be good to have a clear, succinct description of what new feature allowed you to train spiking networks with a local, biologically realistic learning rule, something that others have failed to do. Perhaps it's the auto-encoder? (See next comment.)

4) The auto-encoder is critical for the operation of the network. Can you provide any intuition about why it's there, and what it does? This may not be possible, but it would greatly strengthen the manuscript if you could do it.

5) A potentially important limitation of this paper is that it does not take into account feedback delays during training. This would be a significant limitation if the purpose of this work is to propose how forward models can be learned in the brain: one of the main reasons to have forward models in the first place is being able to deal with feedback delays (see e.g. Wolpert, Ghahramani & Jordan, 1995). Learning that ignores feedback delays is therefore not fully biologically realistic. If this limitation is indeed correct, it needs to be mentioned more prominently in the Discussion section (we only found it buried in the Methods section); if the network can in fact handle feedback delays during training (that are on a similar scale as the target dynamics), it should be demonstrated, as it would make the impact of the paper significantly stronger.

6) An alternative approach to training recurrent networks is to initialize the recurrent weights in some appropriate way and to just train the readout weights during learning (reservoir computing). How much more computationally powerful is the proposed method? A simple way to test this would be to compare the performance of the trained network to that of a network where the recurrent weights have been shuffled after training, the feedback weights set to zero, and the output weights retrained by linear regression. If such a network had comparable computational performance, it would suggest that only the correct recurrent weight distribution is required to learn the task.

7) Does learning still work if the dynamical system has small amounts of stochasticity, or if the error signal is noisy?

8) It's critical to have experimental predictions. Or at least a summary of the implications of this work for what's going on in the brain.

[Editors' note: further revisions were requested prior to acceptance, as described below.]

Thank you for submitting your revised article "Predicting non-linear dynamics by stable local learning in a recurrent spiking neural network" for consideration by *eLife*. The evaluation has been overseen by a Peter Latham as the Reviewing Editor and Timothy Behrens as the Senior Editor.

I want to congratulate you on a very nice paper, and thank you for your very thorough responses; you addressed almost all the points. The only problem is that in parts the manuscript is still not very clear (at least not to me). Please take a very hard look at the Methods section "Network architecture and parameters", and make sure it's crystal clear.

---

## [Author Response]

Summary:This paper proposes a learning scheme that allows a recurrent network of spiking neurons to predict the dynamics of a non-linear system by training feedforward and recurrent connections. The learning is achieved using a local learning rule. The error signal is fed back by fixed, random connections, with the feedback loop forming an auto-encoder. After learning, which is done via motor "babbling," the network is able to reproduce the output in an open-loop setting without feedback. The authors showcase their training algorithm in a number of linear and non-linear applications, including the dynamics of a two-link arm. In addition, the authors prove analytically under mild assumptions that learning converges if the non-linear dynamics can be realized by the network. The paper is mainly clearly written, and has the potential to be of significant interest to the community.

We are sincerely thankful to the reviewers and editors for their insightful comments. We have incorporated all the essential revisions and minor points in our revised manuscript, as detailed in the individual responses below. We hope we have addressed them all to your satisfaction.

Essential revisions:1) The firing rates are far higher than is biologically plausible. It is important that this is fixable, at least in principle. If not, then the scheme has nothing to do with the brain. We hesitate to ask for additional simulations, but we think you need to show that the network can work when firing rates are in a more biologically plausible range, as in less than about 20 Hz on average (still high, but closer to reality).

We thank the reviewers for exhorting us to reduce the firing rates to a more biologically plausible range.

We have replaced Figure 2 with newer simulations run with modified neuronal parameters that yield 13 Hz mean firing rate (averaged over 16s) now compared to 37 Hz before. Third paragraph of subsection “Non-linear oscillator” now reads:

“The distributions of firing rates averaged over a 0.25 s period with fairly constant output, and over a 16 s period with time-varying output, were long-tailed, with the mean across neurons maintained at approximately 12-13 Hz (Figure 2).”

We have also modified the Methods section to include these different neuronal parameters in the second para of subsection “Network architecture and parameters”:

“For the low firing rate simulation in Figure 2, the gains $\nu_i$ were all set to 2, with the biases chosen uniformly from (−2, 2). Choosing a fixed gain for all neurons led to lower variance in firing rates compared to random gains.”

See also Figure 1—figure supplement 1 for representative neuronal tuning curves.

2) We did not follow a key step. The sixth equation in subsection “network architecture and parameters” says:sum_{beta, i} e_{j,beta} d_{beta,i} a_i(sum_alpha e_{i,alpha} epsilon_alpha)\approx sum_alpha e_{j,alpha} epsilon_alphaIt seems that you effectively rewrote this assum_{beta, i} e_{j,beta} d_{beta,i} a_i(I_i) \approx I_i(see Eq. (13)). Strictly speaking, this holds only if the current (whose ith component is I_i) lies in the low dimensional submanifold spanned by the e_{i,alpha}. However, because of the recurrent dynamic, the network is likely be chaotic, and so the current is likely to live on a high-dimensional manifold.This seems to indicate a problem with the derivation. Or are we missing something? In any case, this needs to be explained much more clearly.

We profusely thank the reviewers for pointing out this issue.

The neural input actually remains always in the requisite low-dimensional manifold. We have now clarified and derived it in detail in the text as below.

Below equation (17) for the compressive auto-encoder we write:

“Thus, the loss function of equation (17) is approximately proportional to the squared-error loss function of equation (15) used for the expansive auto-encoder, showing that for an auto-encoder embedded in a loop with fixed random encoding weights, the expansive and compressive descriptions are equivalent for those $N$-dimensional inputs $I_i$ that lie in the $N_d$-dimensional sub-space spanned by $\{e_{i\alpha}\}$ i.e. $I_i$ is of the form $\sum_\alpha e_{i\alpha} \epsilon_\alpha$ where $\epsilon_\alpha$ lies in a finite domain (hypersphere).”

Below equation (25) [above-referred previous equation (13)] in the first major step of the proof, we have added:

“This is valid only if the high-$N$-dimensional input $I_i$ lies in the low-$N_d$-dimensional subspace spanned by $\{e_{i\alpha}\}$ (equation (17)). We show that this requirement is fulfilled in the next major step of the proof (see text accompanying equations (31)-(35)).”

The text in subsection “Error-feedback loop ensures that output follows reference” around new equations (31)-(35) in the second major step of the proof confirms this requirement. The text is too large to excerpt here.

In brief, the derivation shows that the fed-back error current dominates over the feedforward and recurrent contributions, beginning with small feedforward and recurrent weights, at the start of learning. The fed-back error input lies in the low dimensional manifold spanned by the error encoding weights, driving the neural activity close to ideal (also in a low-dimensional manifold) and this in turn ensures that the feedforward and recurrent weights evolve to contribute terms essentially in the low-dimensional manifold. Thus, at all times in the learning and testing, the neural currents lie in a low dimensional manifold.

3) It would be good to have a clear, succinct description of what new feature allowed you to train spiking networks with a local, biologically realistic learning rule, something that others have failed to do. Perhaps it's the auto-encoder? (See next comment.)

We thank the reviewers for asking us to clarify the new features of our learning scheme.

We address both points 3 and 4 with this response. The key new features are the use of the error-feedback loop along with the auto-encoder.

We have introduced a sub-section early in the Results section titled: “Negative-feedback of error via auto-encoder enables local learning”.

This incorporates old and new text, explaining what new features enable learning and how.

In brief, the error feedback using the auto-encoder of gain k, serves two purposes: (1) it makes the projected error available as a local current in every neuron in the recurrent network, and (2) it drives each neuron's activity in the recurrent network to be close to ideal. Thus, since the activities are already ideal, only the weights are responsible for the output error, and the weights can be trained with a simple perceptron-like learning rule, overcoming the credit-assignment problem. In FORCE, for example, neural activities are also forced close to ideal, but by very fast weights swings initially, at a time scale faster than the reference dynamics (biologically implausible), and without the error being projected locally in each neuron (non-local).

In particular please see the paragraphs (much easier to read in the typeset pdf, hence not excerpting fully here):

Subsection “Negative error feedback via auto-encoder enables local learning**”**:

"The combination of the auto-encoder and the error feedback implies that the output stays close to the reference, as explained now..."

In the Methods section, we show that not just the low-dimensional output $\vec{\hat{x}}$, but also the spike trains $S_i(t)$, for $i=1,\ldots,N$, are entrained by the error feedback to be close to the ideal ones required to generate $\vec{x}$.

"During learning, the error feedback via the auto-encoder in a loop serves two roles: (i) to make the error current available in each neuron, projected correctly, for a local synaptic plasticity rule, and (ii) to drive the spike trains to the target ones for producing the reference output. In other learning schemes for recurrent neural networks, where neural activities are not constrained by error feedback, it is difficult to assign credit or blame for the momentarily observed error, because neural activities from the past affect the present output in a recurrent network. In the FOLLOW scheme, the spike trains are constrained to closely follow the ideal time course throughout learning, so that the present error can be attributed directly to the weights, enabling us to change the weights with a simple perceptron-like learning rule \citep{rosenblatt_principles_1961} as in equation (10), bypassing the credit assignment problem. In the perceptron rule, the weight change $\Δ w \sim (\text{pre})\cdot \δ$ is proportional to the presynaptic input $(\text{pre})$ and the error $\δ$. In the FOLLOW learning rule of equation (10), we can identify $(S_i*\kappa)$ with $(\text{pre})$ and $(I_i^\epsilon * \kappa^\epsilon)$ with $\δ$. In Methods, we derive the learning rule of equation (10) in a principled way from a stability criterion."

We further mention how this is different from FORCE and some other similar methods (briefly here after introducing the rule and above intuition in the results, see also the detailed comparisons in the Discussion section):

"FORCE learning [Sussillo and Abbott, 2009; 2012; DePasquale et al., 2016; Thalmeier et al., 2016; Nicola and Clopath, 2016] also clamps the output and neural activities to be close to ideal during learning, by using weight changes that are faster than the time scale of the dynamics. In our FOLLOW scheme, clamping is achieved via negative error feedback using the auto-encoder, which allows weight changes to be slow and makes the error current available locally in the post-synaptic neuron. Other methods used feedback based on adaptive control for learning in recurrent networks of spiking neurons, but were limited to linear systems [MacNeil and Eliasmith, 2011; Bourdoukan and Deneve, 2015], whereas the FOLLOW scheme was derived for non-linear systems (see Methods section). Our learning rule of equation (10) uses an error $\epsilon_\alpha \equiv x_\alpha – \hat{x}_\alpha$ in the observable state, rather than an error involving the derivative $dx_\alpha/dt$ in equation (1), as in other schemes (see Appendix 1) [Eliasmith, 2005; MacNeil and Eliasmith, 2011]. The reader is referred to the Discussion section for detailed further comparisons. The FOLLOW learning rule is local since all quantities needed on the right-hand-side of equation (10) could be available at the location of the synapse in the postsynaptic neuron. For a potential implementation and prediction for error-based synaptic plasticity, and for a critical evaluation of the notion of ‘local rule’, we refer to the Discussion section."

4) The auto-encoder is critical for the operation of the network. Can you provide any intuition about why it's there, and what it does? This may not be possible, but it would greatly strengthen the manuscript if you could do it.

Thanks for asking us to clarify the auto-encoder. It is key to ensuring that the error is fed-back projected correctly to the neurons for a local learning rule, and to ensure that the neurons are driven to the ideal activities by the error current.

The text added for this clarification is incorporated in the response for the above revision.

See in particular Subsection “Negative error feedback via auto-encoder enables local learning**”**.

“The combination of the auto-encoder and the error feedback implies that the output stays close to the reference, as explained now. In open loop i.e. without connecting the output $\vec{\hat{x}}$ and the reference $\vec{x}$ to the error node, an input $\vec{\epsilon}$ to the network generates an output $\vec{\hat{x}} = k\vec{\epsilon}$ due to the auto-encoder of gain $k$. In closed loop, i.e. with the output and reference connected to the error node (Figure 1), the error input is $\vec{\epsilon} = \vec{x}-\vec{\hat{x}}$, and the network output $\vec{\hat{x}}$ settles to:

\vec{\hat{x}} = k\vec{\epsilon} = k\left(\vec{x}-\vec{\hat{x}}\right)

\implies \vec{\hat{x}} = \frac{k}{k^+^1}\vec{x} \approx \vec{x},

i.e. approximately the reference $\vec{x}$ for large positive $k$. The fed-back residual error $\vec{\epsilon}=\vec{x}/(k^+^1)$ drives the neural activities and thence the network output. Thus, feedback of the error causes the output $\hat{x}_\alpha$ to approximately follow $x_\alpha$, for each component $\alpha$, as long as the error feedback time scale is fast compared to the reference dynamical system time scale, analogous to negative error feedback in adaptive control [Narendra and Annaswamy, 1989; Ioannou and Sun, 2012].”

5) A potentially important limitation of this paper is that it does not take into account feedback delays during training. This would be a significant limitation if the purpose of this work is to propose how forward models can be learned in the brain: one of the main reasons to have forward models in the first place is being able to deal with feedback delays (see e.g. Wolpert, Ghahramani & Jordan, 1995). Learning that ignores feedback delays is therefore not fully biologically realistic. If this limitation is indeed correct, it needs to be mentioned more prominently in the Discussion section (we only found it buried in the Methods section); if the network can in fact handle feedback delays during training (that are on a similar scale as the target dynamics), it should be demonstrated, as it would make the impact of the paper significantly stronger.

Thank you for pointing out this important use case of the forward model vis-a-vis our learning scheme.

We performed these simulations and incorporated the results (Figure 6). While our learning scheme cannot really learn a dynamical system followed by a delay taken together as the reference system, we propose a scheme in the Discussion section (Figure 7) using the Smith predictor configuration, adapted from suggestions by Miall & Wolpert, 1996 and Smith, 1957, whereby our forward model can learn with and compensate for sensory feedback delays in motor control. Our forward model is able to feedback a corrective prediction to the controller, before the sensory feedback arrives, thus improving control.

In the Results section, we have added panels E-H in Figure 6 and added further text at the very end of Results section:

"We also asked if the network could handle sensory feedback delays in the reference signal. Due to the strong limit cycle attractor of the van der Pol oscillator, the effect of delay is less transparent than for the linear decaying oscillator (Figure 2—figure supplement 2), so we decided to focus on the latter. For the linear decaying oscillator, we found that learning degraded rapidly with a few milliseconds of delay in the reference, i.e. if $x(t − \Δ)$ was provided as reference instead of $x(t)$ (Figure 6). We compensated for the sensory feedback delay by delaying the motor command input by identical $\Δ$ (Figure 6), which is equivalent to time-translating the complete learning protocol, to which the learning is invariant, and thus the network would learn for arbitrary delay (Figure 6). In the Discussion section, we suggest how a forward model learned with a compensatory delay (Figure 6) could be used in control mode to compensate for sensory feedback delays."

In the Discussion section, we have added panels B and C in Figure 7 and the text below:

“One of the postulated uses of the forward predictive model is to compensate for delay in the sensory feedback during motor control [Miall and Wolpert, 1996; Wolpert et al., 1995] using the Smith predictor configuration [Smith, 1957]. We speculate that the switch from the closed-loop learning of forward model with feedback gain k >> 1 to open-loop motor prediction k = 0 could also be used to switch delay lines: the system can have either a delay before the forward model as required for learning (Figure 7), or after the forward model as required for the Smith predictor (Figure 7). We envisage that FOLLOW learning of the forward model occurs in closed loop mode (k >> 1) with a delay in the motor command path, as outlined earlier in Figure 6 and now embedded in the Smith predictor architecture in Figure 7. After learning, the network is switched to motor control mode, with the forward predictive model in open loop (k = 0), implementing the Smith predictor (Figure 7). In this motor control mode, the motor command is fed with zero delay to the forward model. This enables to rapidly feed the estimated state back to the motor controller so as to take corrective actions, even before sensory feedback arrives. In parallel, available sensory feedback is compared with a copy of the forward model that has passed through a compensatory delay after the forward model (Figure 7).”

6) An alternative approach to training recurrent networks is to initialize the recurrent weights in some appropriate way and to just train the readout weights during learning (reservoir computing). How much more computationally powerful is the proposed method? A simple way to test this would be to compare the performance of the trained network to that of a network where the recurrent weights have been shuffled after training, the feedback weights set to zero, and the output weights retrained by linear regression. If such a network had comparable computational performance, it would suggest that only the correct recurrent weight distribution is required to learn the task.

Thank you for this interesting comparison to reservoir computing.

We performed these simulations. Our recurrent network does not behave like a reservoir. Rather as shown for revision 2, the neural activities remain low-dimensional as they are driven strongly by the error current.

At the end of subsection “Non-linear oscillator” (van der Pol) oscillator, we added the text below:

“We also asked whether merely the distribution of the learned weights in the recurrent layer was sufficient to perform the task, or whether the specific learned weight matrix was required. This question was inspired from reservoir computing [Jaeger, 2001; Maass et al., 2002; Legenstein et al., 2003; Maass and Markram, 2004; Jaeger and Haas, 2004; Joshi and Maass, 2005; Legenstein and Maass, 2007], where the recurrent weights are random, and only the readout weights are learned. To answer this question, we implemented a perceptron learning rule on the readout weights initialized at zero, with the learned network’s output as the target, after setting the feedforward and / or recurrent weights to either the learned weights as is or after shuffling them. The readout weights could be approximately learned only for the network having the learned weights and not the shuffled ones (Figure 2—figure supplement 3), supporting the view that the network does not behave like a reservoir (Methods section).”

We have also added the corresponding text in the Methods section.

7) Does learning still work if the dynamical system has small amounts of stochasticity, or if the error signal is noisy?

Thanks for asking about noise in the reference / error signals. Yes, it is robust.

We have added in the last sub-section of Results section, panel B in Figure 6 with accompanying text:

“We added Gaussian white noise to each component of the error, which is equivalent to adding it to each component of the reference, and ran the van der Pol oscillator learning protocol for 10,000 s for different standard deviations of the noise (Figure 6). The learning was robust to noise with standard deviation up to around 0.001, which must be compared with the error amplitude of the order of 0.1 at the start of learning, and orders of magnitude lower later.”

8) It's critical to have experimental predictions. Or at least a summary of the implications of this work for what's going on in the brain.

Thank you for pointing out this important lacuna. Our two predictions about error-current driven synaptic plasticity and the presence of an auto-encoder in a feedback loop have been highlighted now.

We have added in the Discussion section with Figure 7 prediction about error-current driven synaptic plasticity:

“A possible implementation in a spatially extended neuron would be to imagine that the post-synaptic error current $I_i^\epsilon$ arrives in the apical dendrite where it stimulates messenger molecules that quickly diffuse or are actively transported into the soma and basal dendrites where synapses from feedforward and feedback input could be located, as depicted in Figure 7. Consistent with the picture of a messenger molecule, we low-pass filtered the error current with an exponential filter $\kappa^\epsilon$ of time constant 80 ms or 200 ms, much longer than the synaptic time constant of 20 ms of the filter $\kappa$. Simultaneously, filtered information about presynaptic spike arrival $S_j*\kappa$ is available at each synapse, possibly in the form of glutamate bound to the postsynaptic receptor or by calcium triggered signalling chains localized in the postsynaptic spines. Thus, the combination of effects caused by presynaptic spike arrival and error information available in the postsynaptic cell drives weight changes, in loose analogy to standard Hebbian learning.

The separation of the error current from the currents at feedforward or recurrent synapses could be spatial (such as suggested in Figure 7) or chemical if the error current projects onto synapses that trigger a signalling cascade that is different from that at other synapses. Importantly, whether it is a spatial or chemical separation, the signals triggered by the error currents need to be available throughout the postsynaptic neuron. This leads us to a prediction regarding synaptic plasticity that, say in cortical pyramidal neurons, the plasticity of synapses that are driven by pre-synaptic input in the basal dendrites, should be modulated by currents injected in the apical dendrite or on stimulation of feedback connections.”

Further, the presence of an auto-encoder in a feedback loop is a prediction also added to the Discussion section:

“The first assumption is that error encoding feedback weights and output decoding readout weights form an auto-encoder. This requirement can be met if, at an early developmental stage, either both sets of weights are learned using say mirrored STDP [Burbank, 2015], or the output readout weights are learned, starting with random encoding weights, via a biological perceptron-like learning rule [D'Souza et al., 2010; Urbanczik and Senn, 2014]. A pre-learned auto-encoder in a high-gain negative feedback loop is in fact a specific prediction of our learning scheme, to be tested in systems-level experiments.”

[Editors' note: further revisions were requested prior to acceptance, as described below.]

Thank you for submitting your revised article "Predicting non-linear dynamics by stable local learning in a recurrent spiking neural network" for consideration by eLife. The evaluation has been overseen by a Peter Latham as the Reviewing Editor and Timothy Behrens as the Senior Editor.I want to congratulate you on a very nice paper, and thank you for your very thorough responses; you addressed almost all the points. The only problem is that in parts the manuscript is still not very clear (at least not to me). Please take a very hard look at the Methods section "Network architecture and parameters", and make sure it's crystal clear.

We thank you for the compliments and for pointing out the lack of clarity in the Methods subsection: "Network architecture and parameters". We have re-written the unclear subsection, and we hope that the same is now crystal clear. The changes are outlined below.

We have substantially re-written the said Methods subsection. It is now renamed "Network parameters" as the network architecture was already described at the start of the Results section. We have partitioned this subsection into smaller subheadings. The key changes are, in brief, that the gains and biases of neurons and the encoding weights for each layer are first clearly defined. We then explain the procedure for initializing them randomly at the start of a simulation, after which they remain fixed. Figure —figure supplement 1 has also been suitably modified to show the different gain functions of the heterogeneous neurons. We have also improved the explanation of setting the readout weights to form an auto-encoder with respect to the error encoding weights, as well as improved the discussion of the expansive versus compressive view of our auto-encoder.

We have also made a few minor word-level corrections / improvements in other sections.